# Nonlinear El Niño impacts on the global economy under climate change

Yi Liu [1,2], Wenju Cai [1,2,3,4] ✉, Xiaopei Lin [1,3], Ziguang Li[1,3] & Ying Zhang[5] ✉

The El Niño-Southern Oscillation (ENSO) is a consequential climate phenomenon affecting global extreme weather events often with largescale socio-economic impacts. To what extent the impact affects the macroeconomy, how long the impact lasts, and how the impact may change in a warming climate are important questions for the field. Using a smooth nonlinear climate-economy model fitted with historical data, here we find a damaging impact from an El Niño which increases for a further three years after initial shock, amounting to multi-trillion US dollars in economic loss; we attribute a loss of US$2.1 T and US$3.9 T globally to the 1997-98 and 2015-16 extreme El Niño events, far greater than that based on tangible losses. We find impacts from La Niña are asymmetric and weaker, and estimate a gain of only US$0.06 T from the 1998-99 extreme La Niña event. Under climate change, economic loss grows exponentially with increased ENSO variability. Under a high-emission scenario, increased ENSO variability causes an additional median loss of US$33 T to the global economy at a 3% discount rate aggregated over the remainder of the 21st century. Thus, exacerbated economic damage from changing ENSO in a warming climate should be considered in assessments of mitigation strategies.

The El Niño-Southern Oscillation (ENSO) is a consequential climate phenomenon on Earth, alternating between the warm phase El Niño and cool phase La Niña[1,2]. During El Niño, anomalous sea surface temperature (SST) warming in the central and eastern equatorial Pacific weakens the easterly trade winds and shifts the atmospheric deep convection eastward, which in turn alters global atmospheric circulation through its teleconnections[3–6]. During La Niña, the reverse generally occurs but not symmetrically. The impact has a global reach, affecting a myriad of fields transcending extreme weathers, hydrological cycle, ecosystems and agriculture to human community[7–10]. In contrast to substantial advances in our understanding of ENSO's physical dynamics and ENSO global climate teleconnection, how ENSO affects human society and how the impact might change under climate change remain important areas to explore.

Major El Niño and La Niña events are known to cause substantial economy reductions in affected countries and regions, directly through local weather extremes[11–13]: for example, the 1982-83 El Niño caused severe floods in southeast Brazil, with economic losses exceeding US$1.1 billion (ref. 14); the 1997-98 El Niño led to extreme weathers around the United States and direct economic losses of ~US$4 billion (ref. 15); the 1998-99 La Niña contributed to one of the most devastating Yangtze floods in China, with direct economic losses amounted to US$20 billion (ref. 16); the 2015-16 El Niño resulted in multi-year extreme drought and wildfire in Amazonia that cost about US$26 billion in total[17]. There are well-established connections between ENSO and subcomponents of economic production, such as crop yields and fisheries[18,19], and losses at micro levels are reflected on macro levels[20,21]. However, the effect from El Niño and La Niña do not compensate[22,23].

Climate change affects many aspects of modern human society including economy, public health and human conflict[24–27]. The mean temperature and precipitation have been common metrics for

[1]Physical Oceanography Laboratory/Frontiers Science Center for Deep Ocean Multispheres and Earth System/Sanya Oceanographic Institution, Ocean University of China, Qingdao, China. [2]CSIRO Environment, Hobart, Tasmania, Australia. [3]Laoshan Laboratory, Qingdao, China. [4]State Key Laboratory of Loess and Quaternary Geology, Institute of Earth Environment, Chinese Academy of Sciences, Xi'an, China. [5]School of Management, Ocean University of China, Qingdao, China. ✉e-mail: Wenju.Cai@csiro.au; yzhang@ouc.edu.cn

assessing the climate change impact. Recent advances suggest that ENSO variability is likely to increase under greenhouse warming[28,29]. Whether changes in ENSO cycles intensify the economic risks of greenhouse warming is an important issue. Here, by assessing ENSO's impact on global economic production using historical climate and economic production data, we find that, in contrast to an nondefinitive effect from La Niña, there is a substantial negative impact from El Niño on global economic production that lasts for multi-years, leading to a larger growth effect than previously thought; further, the impact from future ENSO cycles increases in a warming climate, with an exacerbated loss of global economic production from increased ENSO amplitude under greenhouse warming.

## Results and discussion

### Nonlinear effect of ENSO on economic production

Most empirical econometric models treat ENSO as a simple linear predictor and focus on the negative impact of El Niño[20,30,31]. This approach assumes that La Niña has a positive impact symmetric to negative impact of El Niño. However, a limited nonlinear model incorporating a Heaviside step function to depict impact of La Niña and El Niño deferentially[22,23] finds that La Niña may not always benefits; for example, the 1998-99 strong La Niña causes extreme rainfall and floods that reduce economic growth and damage livelihoods in many affected countries[16,32], which substantially reduce the beneficial effect of a La Niña. This leads to heterogeneous economic effects across countries (Supplementary Fig. S1), and any benefit may be offset or overwhelmed by the losses. Further, ENSO is a nonlinear-dynamical system with asymmetric amplitude of SST anomalies and teleconnections between El Niño and La Niña[33,34]; for example, El Niño amplitude tends to be greater than that of La Niña, exerting a greater impact on global climate and human community than La Niña, and such impact can increase or change nonlinearly with amplitude of El Niño/La Niña. In addition, impact of an ENSO event could affect economic production in the ensuing years after global economic connectivity is disrupted by the ENSO event[22]. Therefore, an econometric model must reflect these features to realistically estimate ENSO's effect on global economy.

As such, we establish a fixed-effect panel regression model[35] using an ENSO index as a nonlinear predictor and incorporating lagged effects (see 'Empirical econometric model' in Methods). The ENSO index is Niño3.4 SST anomaly (averaged over 5°S–5°N, 120°–170°W) in boreal winter (December, January and February, D(0)JF(1)), when ENSO typically peaks. The model accounts for: (1) country-fixed effect, such as different history and culture backgrounds of individual countries; (2) country-specific long-term linear and quadratic time trends in growth rates, derived from changing political institutions and economic policies of individual countries; and (3) nonlinear effect of annual mean country-specific temperature and precipitation, as in previous studies[25,36], except that ENSO signal is removed through linear regression to obtain ENSO-independent annual temperature and precipitation. Importantly, we incorporate the nonlinear effect of ENSO in a quadratic function in which both the linear and the nonlinear components include lagged effects. These lag terms account for growth effects after the contemporaneous climate shock of an ENSO event.

We train our model using climate and economic data over 1960–2019 period. Time-invariant and time-trending covariates are allowed to interact with observed economic and climate variables. We include the mean temperature and precipitation terms to parallelly estimate the effect from mean state change and from ENSO. Consistent with previous findings[25,36], the nonlinear effect of mean temperature operates in addition to the ENSO impact. In contrast to the mean temperature which has a significant contemporaneous effect[25,36], we find that ENSO impact persists for 3 years (Supplementary Fig. S2), after which time little further impact is seen and uncertainty increases.

We therefore focus on its effect with lags of 3 years after an event. A set of Bootstrap methods is applied to quantify uncertainty of point estimates in the model (Supplementary Fig. S3; see 'Statistical significance test' in Methods).

Our model reveals a nonlinear relationship between the Niño3.4 index and its economic impact on country-level economic production (Fig. 1a). Both extreme El Niño and La Niña cause damage on economic growth, but the damage is far greater during El Niño than during La Niña; weak and moderate La Niña events produce a smaller benefit, which in amplitude is far smaller than the damage of weak El Niño events. Overall, there is a negative and statistically significant impact on economic growth during El Niño, but the impact is by and large insignificant for La Niña. Our model also finds that the contemporaneous effect is dominated by the linear component and its growth effect is dominated by the nonlinear component (Supplementary Table S1); the difference between the impact in the occurrence year and in the lagged years reflects a subsequent acceleration in the impact on economic activity that is fed by the initial shock.

We test the heterogeneity of such nonlinear effect by different groups of countries. Firstly, we measure monthly ENSO teleconnection to each country at different lags and accumulate teleconnections that are statistically significant to separate teleconnected and weakly-teleconnected countries (Supplementary Fig. S5–6; see 'ENSO index and country-specific teleconnection'). ENSO-teleconnected countries exhibit a greater response to El Niño than weakly-teleconnected countries (Fig. 1b), because the economic impacts from ENSO are underpinned by the direct response of local weather and climate conditions[23]. We choose to involve all countries in our main analysis to account for indirect influences like trade. Secondly, agriculture-dependent countries, of which the GDP share of agriculture is >20%, show a greater response to El Niño than agriculture-independent countries because agriculture production is more likely to be affected by ENSO-induced local weather anomalies (Fig. 1c). Similarly, lower-income countries, that is, with a purchasing-power-parity-adjusted Gross Domestic Product (GDP) per capita in 1980 below the median[25], exhibit a greater response to El Niño than high-income countries because their level of preparedness and the capacity to mitigate are relatively low, they are mostly located in the tropics where the ENSO teleconnections are strong, and their economies tend to have a high proportion of agriculture (Fig. 1d). The common nonlinear nature reflects the feature that the global spillovers and cascading effects dominate the global economic impact of ENSO (see 'Heterogeneity of response function' in Methods).

### Economic loss an order of magnitude greater than previously thought

In contrast to previous estimates[37,38] that the extreme El Niño events of 1982–83, 1997−98 and 2015−16 each costed global economy tens of billions in US dollars, our result suggests an impact from each event is trillions of US dollars. These additional losses mainly come from unobservable and delayed response to the initial climate shock. Initially, El Niño drives direct losses in the severely-affected regions through extreme weathers as reflected in temperature and precipitation anomalies. Subsequently, cross-sector and cross-border spillovers occur, affecting global macroeconomy. Previous studies suggested the cascading effects may commence via several transmission channels. For example, extreme weathers lower crop yields and agricultural productivity, leading to food shortages, trade contractions and commodity price increases[39]; abnormal sea surface temperature and ocean current cause decreased fishery stocks and other marine resources[40]; damages on infrastructure increase reconstruction and maintenance costs and disrupt transportation networks[41]; fluctuation in rainfall and surface run-off cause hydroelectric power shortages and reduce water-dependent industrial outputs[42]; changes in disease vector dynamics

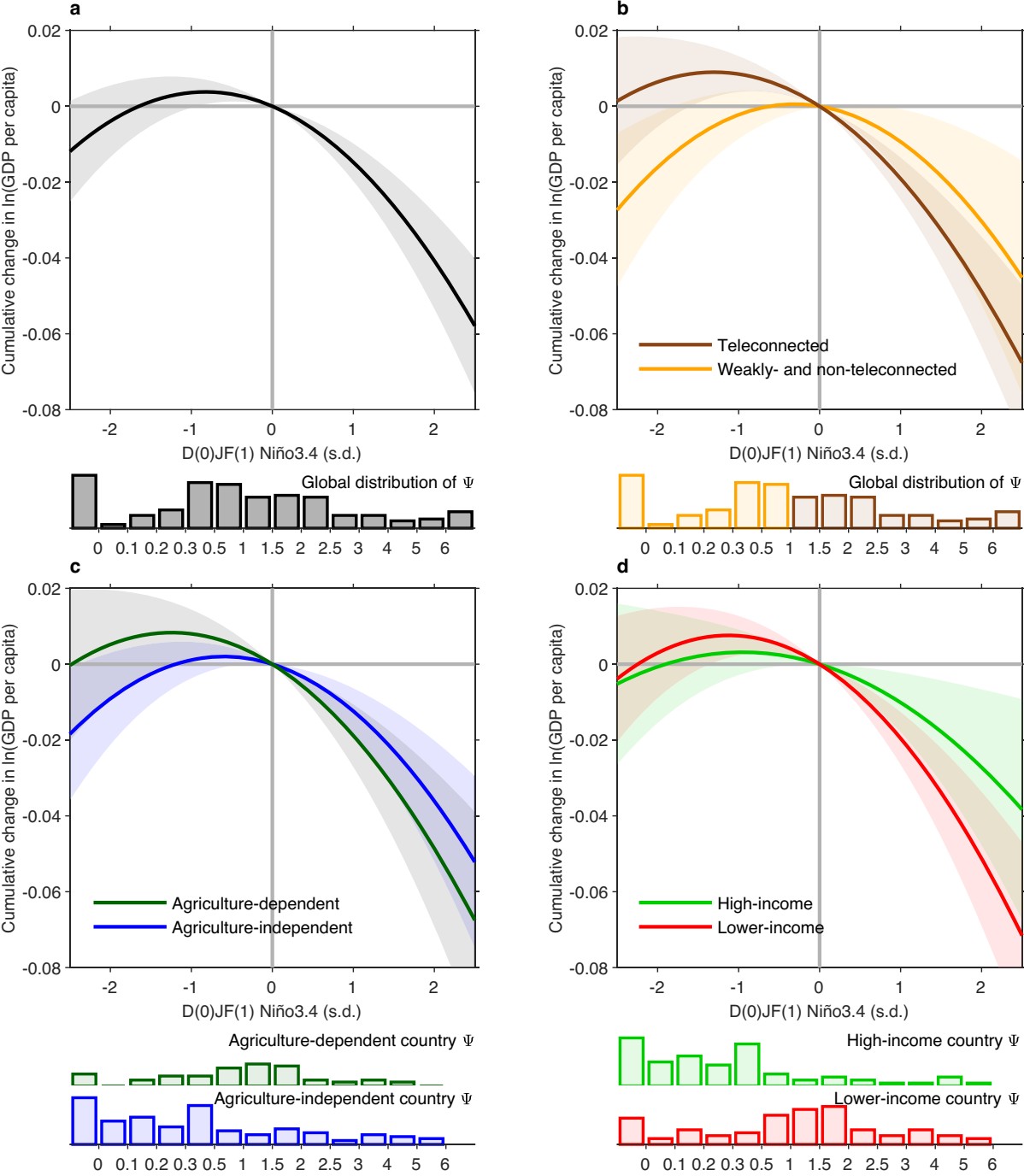

**Fig. 1 | Nonlinear effect of ENSO on global economic production. a** Global nonlinear relationship between D(0)JF(1) Niño3.4 index (normalized) and 3-year cumulative (from year 0 to year 3) change in log GDP per capita for all countries during 1960–2019, with shading indicating the 95% confidence level based on a Bootstrap method (see 'Statistical significance test' in Methods). Model includes country-fixed effects, country-specific linear and quadratic trends, mean temperature and precipitation controls. Histograms below show the distribution of country-specific ENSO teleconnection strength. **b** Same as **a**, but with countries divided into teleconnected group (brown curve) and weakly-teleconnected group (yellow curve). Histograms below show the distribution of country-specific ENSO teleconnection strength for teleconnected (brown bars) and weakly-teleconnected (yellow bars) countries, respectively. **c**, **d** Same as **b**, but for **c** agriculture-dependent country group (green curve) and agriculture-independent country group (blue curve), and **d** high-income country group (green curve) and lower-income country group (red curve).

caused by extreme weathers increase healthcare cost and reduce manual productivity[43]; and poor weather conditions reduce tourist arrivals and consumptions for tourism-dependent regions[44].

Observed extreme El Niño events have considerable growth effect on global economy. The 1982–83, 1997–98 and 2015–16 events decreased growth by 1.0% in the occurrence year, but >5.0% cumulated

over the subsequent 3 years (Supplementary Fig. S7). We estimate the loss value by multiplying the reduction in global GDP growth rate at each lag with the GDP value of the preceding year. The contemporaneous loss amounts to US$246, US$401 and US$739 billion for the extreme El Niño events of 1982–83, 1997–98 and 2015–16, respectively, (about 0.9–1.0% of global GDP at the time) in the El Niño

occurrence year, and the cumulative loss over the occurrence year and the subsequent 3 years reaches US$1.3, US$2.1 and US$3.9 trillion (about 4–5% of global GDP at the time), respectively (Fig. 2a). In the subsequent 3 years, statistical uncertainty increases somewhat as the year advances, but the cumulative loss increases. The greater loss from the 2015–16 and the 1997–98 event than from the 1982–83 El Niño is due to a larger global economy. The contemporaneous loss is dominated by the linear effect and is only a small part of the total loss; the loss from growth effect is dominated by the nonlinear impact and is larger. By contrast, extreme La Niña events such as in 1988-89, 1998-99 and 2010-11 contribute to a fluctuating impact around zero, with a contemporaneous benefit being offset by a negative growth effect. For example, the cumulative gain of 1998-99 La Niña only amounts to US $0.06 trillion (Fig. 2b).

For a given year $t$, the effect of an ENSO event includes lagged impacts from ENSO events of the prior 3 years plus a contemporaneous impact of the ENSO at year $t$. For example, loss of global economic production in 2000 ($t$ = year 2000) includes the contemporaneous and growth effect from 1997-98 strong El Niño and the ensuing 1998-1999 strong La Niña at different lags, as well as ENSO's

own contemporaneous effect in year 2000. The impact on growth rate is computed for all years over the 1960–2019 period (Fig. 2c). Owning to the asymmetric impact between El Niño and La Niña, there is a net reduction in global GDP growth rate in most years. Over the period of 1960–2019, there is an average reduction of 0.6% per annum in global GDP growth rate from ENSO cycles; the cumulative loss in economic production is US$13.5 trillion in total. Extreme El Niños in 1982–83, 1997–98 and 2015–16 account for 54% of the total loss over the 60 years.

We examine likely impacts of country-level heterogeneity by incorporating potential interaction of the common ENSO shock (as in Niño3.4) with country-specific teleconnection (see 'Heterogeneity of response function' in Methods). Although the teleconnection strength provides some heterogeneity between individual countries (Supplementary Fig. S8a), a lack of statistical significance of interaction terms indicates that most of economic impact is reflected in a common shock after the spillovers and cascading effects are transmitted to the global macroeconomy (Supplementary Table S3). As such, including the country-level heterogeneity leads to a similar estimate of global economic loss (Supplementary Fig. S8b).

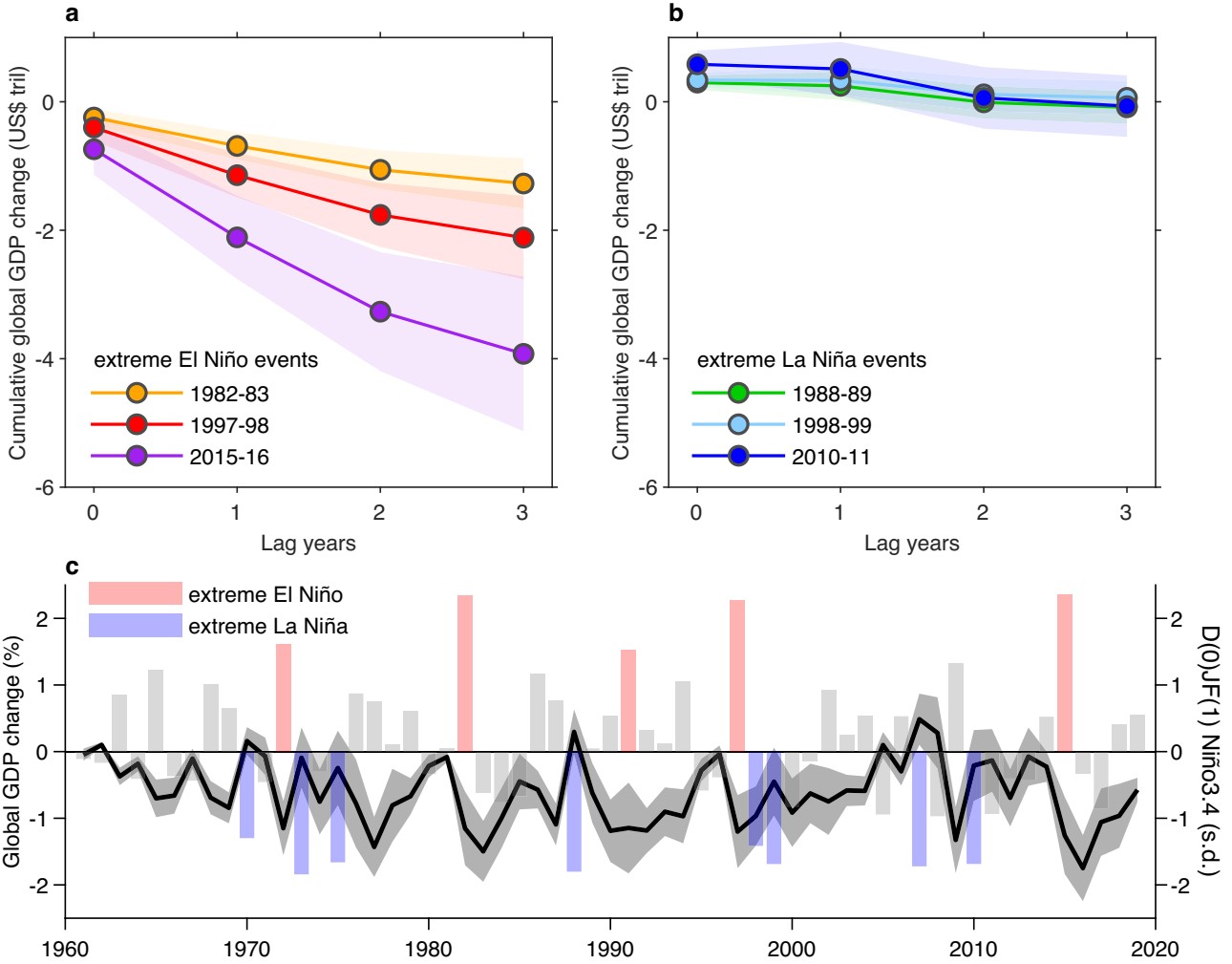

**Fig. 2 | Observed economic production loss from ENSO. a** Cumulative effect of three major extreme El Niño events in 1982/83 (yellow), 1997/98 (red) and 2015/16 (purple) on the global total GDP, accumulating from year 0 to year 3 after the event peaking in D(0)JF(1). Shadings indicate the 95% confidence level for each event based on a Bootstrap method (see 'Statistical significance test' in Methods). **b** Same as **a**, but for three major extreme La Niña events in 1988/89 (green), 1998/99 (sky blue) and 2010/11 (blue). **c** Timeseries of global GDP percentage change from ENSO (black line) and D(0)JF(1) Niño3.4 SST anomaly (normalized, bars). Change in GDP growth rate in each year is calculated as the sum of contemporaneous effect of ENSO event in year 0 and growth effects from year −1 to year −3. Grey shading shows the 95% confidence interval based on a Bootstrap method (see 'Statistical significance test' in Methods). Extreme El Niño (Niño3.4 > 1.5 s.d.) and La Niña (Niño3.4 < −1.25 s.d.) events are marked as red and blue bars, respectively.

## Loss in economic growth increases with ENSO amplitude

We use our econometric model to examine impact of change in future ENSO on global economy under emission scenarios of the Inter-governmental Panel on Climate Change (IPCC). We analyze outputs of available climate models participating in the Coupled Model Inter-comparison Project phase 6 (CMIP6) (ref. 45). These models are forced with historical anthropogenic and natural forcing until 2014, and four Shared Socioeconomic Pathways (SSP) (ref. 46) of future greenhouse gas concentration trajectories from 2015 onwards (see 'Climate and economic data' in Methods).

To depict ENSO evolution in climate models, monthly SST anomalies referenced to the 1900–1999 climatology are constructed and then averaged over the Niño3.4 region. The Niño3.4 ENSO index is then quadratically-detrended over the entire period of 1900–2099. We compare standard deviation of the detrended Niño3.4 index over the 100-year period of 2000–2099 (the 21st century) with that over the 100-years of the 1900–1999 period (the 20th century) to assess ENSO change under greenhouse warming. All four emission scenarios simulate an increase in future ENSO variability with a strong inter-model consensus as reported by previous studies[28,29] (Fig. 3a; Supplementary Fig. S9). For example, in the high-emission scenario of SSP5-8.5, the multi-model ensemble median (mean) increase in future Niño3.4 variability is 14.3% (15.5%). A total of 42 out of 48 available models (87.5%) generate an increase in ENSO variability (Supplementary Fig. S9a). The increased ENSO amplitude is also seen in the SSP3-7.0, SSP2-4.5 and SSP1-2.6 scenarios, with multi-model ensemble median (mean) increase in ENSO amplitude is 13.1%, 9.5%, and 7.5% (15.3%, 11.8%, and 10.6%), respectively (Supplementary Fig. S9b–d).

Conceivably, sequence of ENSO events might affect the ENSO's impact on economy averaged over a period. For example, if an extreme El Niño occurs in the last year of the period, therefore its growth effect is felt beyond the period, it might affect the period-averaged impact. To avoid any potential dependence on ENSO event sequence, we develop a counterfactual 21st century Niño3.4 timeseries

such that it follows the same projected evolution (therefore the same sequence) but its standard deviation is scaled to have the same amplitude as that in the 20th century (Supplementary Fig. S10a; see 'Counterfactual ENSO and scenario' in Methods). The counterfactual future ENSO timeseries is taken as the ENSO evolution if future ENSO variability does not change under greenhouse warming. We then compare ENSO impact on economy using our econometric model with the projected and the counterfactual future ENSO timeseries as inputs. We begin our assessment of the economic impact by the difference in annual global GDP growth rate averaged over the 21st century. This is carried out for each model.

All climate models simulate a net economic growth reduction, and the net reduction becomes greater as the ENSO amplitude increases under all four IPCC emission scenarios, supported by a strong inter-model consensus (Fig. 3b). The multi-model ensemble median (mean) increase in the century-average reduction of global GDP growth is estimated to be 0.19%, 0.18%, 0.12%, and 0.10% (0.25%, 0.23%, 0.19%, and 0.17%) per annum under the SSP5-8.5, SSP3-7.0, SSP2-4.5, and SSP1-2.6 scenario, respectively (Fig. 3b). Further, there is a strong inter-model relationship between changes in ENSO SST variability and changes in century-averaged global GDP growth rate reduction, using all CMIP6 models under all four IPCC emission scenarios (Fig. 3c). Models that generate a greater increase in ENSO amplitude system-atically generate a greater global GDP growth rate reduction.

## Substantial additional loss from future increase in ENSO amplitude

We estimate the ENSO-induced economic production loss from the future growth rate reduction, using projections from the SSPs (ref. 46) that define the secular evolution of country-level population and economic development. The SSP projections cover the period from 2010 to the end of 21st century, forced under different emission levels (see 'Climate and economic data' in Methods). Assuming that loss from ENSO permanently imprints on the long-term economic development,

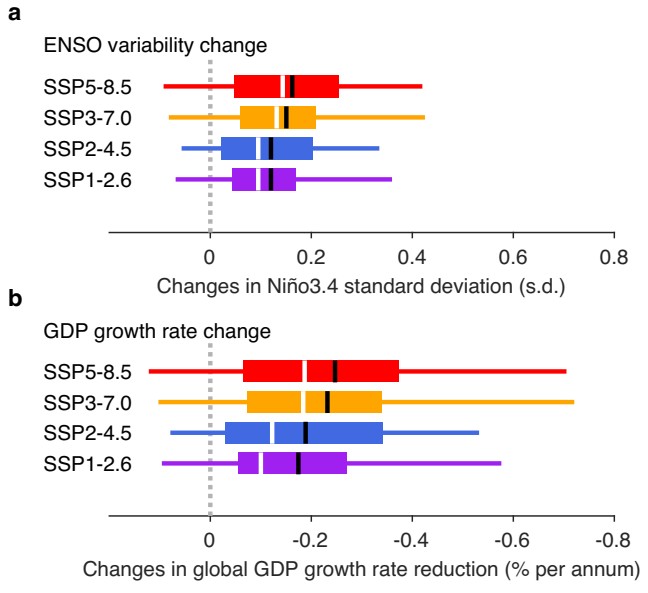

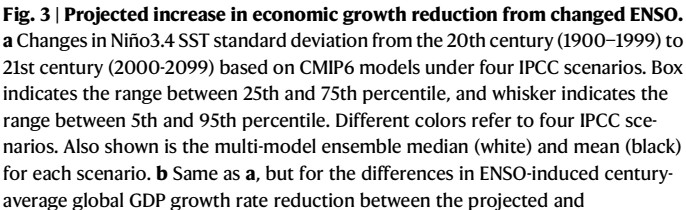

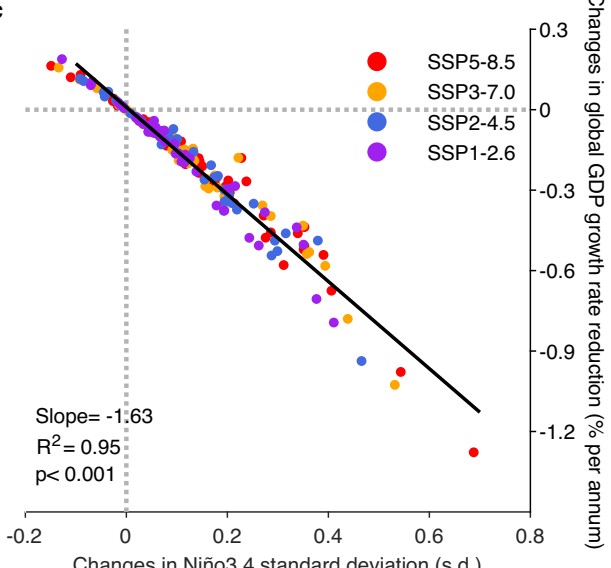

**Fig. 3 | Projected increase in economic growth reduction from changed ENSO.**
**a** Changes in Niño3.4 SST standard deviation from the 20th century (1900–1999) to 21st century (2000-2099) based on CMIP6 models under four IPCC scenarios. Box indicates the range between 25th and 75th percentile, and whisker indicates the range between 5th and 95th percentile. Different colors refer to four IPCC scenarios. Also shown is the multi-model ensemble median (white) and mean (black) for each scenario. **b** Same as **a**, but for the differences in ENSO-induced century-average global GDP growth rate reduction between the projected and

counterfactual scenario in the 21st century from CMIP6 models under four IPCC scenarios. **c** Relationship between changes of Niño3.4 variability (21st century minus 20th century) and differences in ENSO-induced century-average global GDP growth rate reduction (projected minus counterfactual scenario) from CMIP6 models under four IPCC scenarios. Different colors refer to four IPCC scenarios. The slope indicates that, for example, a 1.0 s.d. increase in ENSO amplitude can cause a 1.63% more loss in GDP growth rate. The R square and P-value of fitting are also given.

we develop a counterfactual ENSO for future assuming no change (see 'Counterfactual ENSO and scenario' in Methods), and two "no-ENSO" future economic growth projections from 2020 onward, in which global economic production loss from the projected and the counterfactual ENSO in each year is taken out from the projected GDP timeseries (Supplementary Fig. S10b). The difference is taken as the additional economic loss from changing ENSO. Several fixed-rate discounting schemes, ranging from 1% to 5% per annum, where, for example, a 1% discount rate means that society values a given amount

of consumption in 1 year roughly 1% less than its values today, are employed.

To illustrate, we use distributions of the additional economic loss to quantify the probability under different emission scenarios at a 3% discount rate (Fig. 4a–d). There is >80% chance of an additional economic loss from ENSO changes in all four IPCC emission scenarios, with a median increase ranging from US$14 trillion for the SSP2-4.5 scenario (Fig. 4c) to US$33 trillion for the SSP5-8.5 scenario (Fig. 4a), aggregated over the last 80 years of the 21st century.

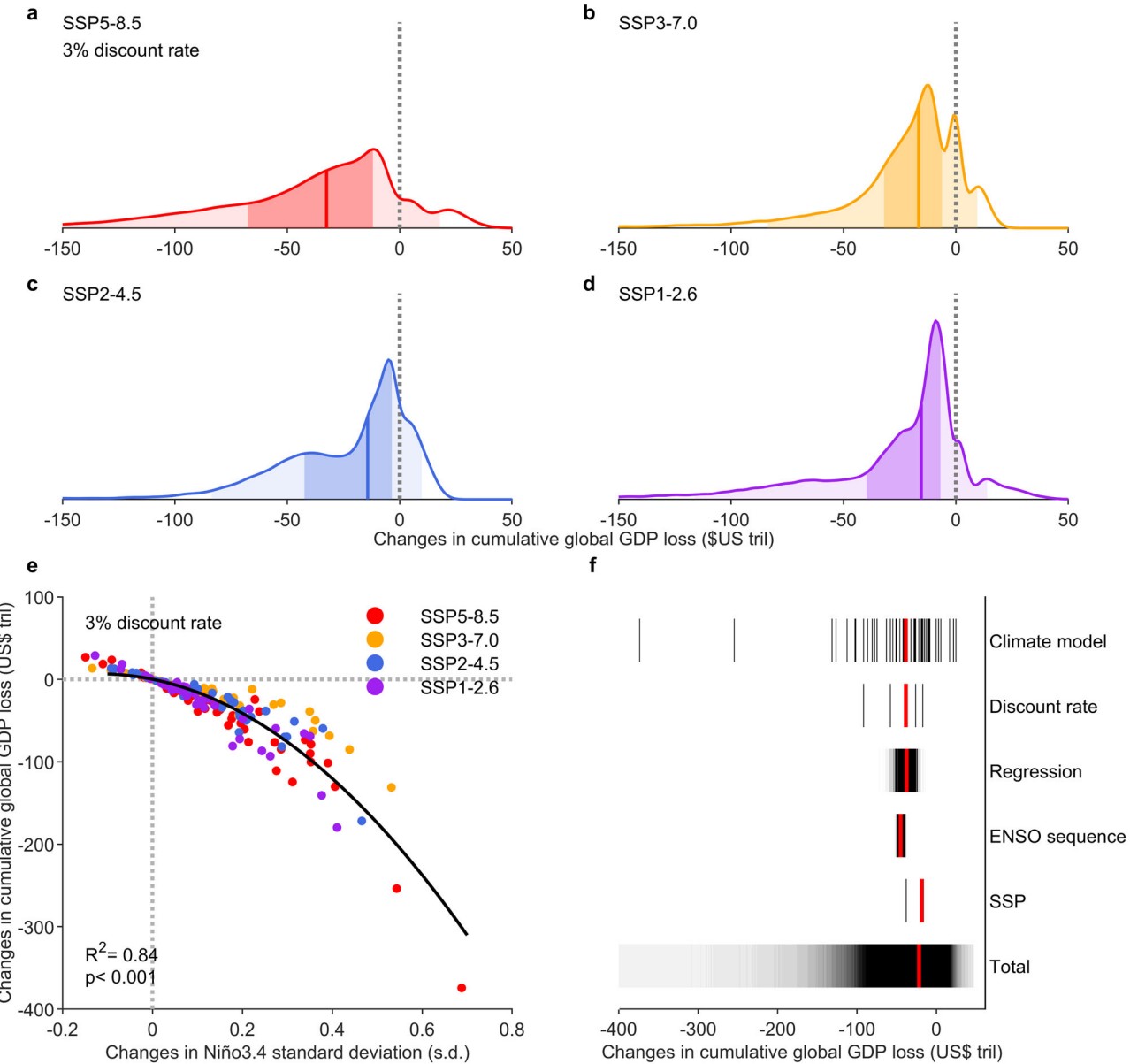

**Fig. 4 | Projected increase in economic production loss from changed ENSO. a–d** Probability distribution of the future change in ENSO-induced economic production loss under **a** SSP5-8.5, **b** SSP3-7.0, **c** SSP2-4.5, and **d** SSP1-2.6 scenario at a 3% discount rate, aggregated over the period of 2020–2099. Light and dark shading for each distribution indicate the 5th–95th and 25th–75th range, respectively, with solid vertical line showing the median. **e** Nonlinear relationship between changes of Niño3.4 variability (21st century minus 20th century) and changes of ENSO-induced cumulative global GDP loss (projected minus counterfactual scenario) over the period of 2020–2099 from CMIP6 models under four IPCC scenarios. Different colors refer to four IPCC scenarios. The $R$ square and $P$-value of fitting are given. **f** Sources of uncertainty of the estimated changes of the cumulative global GDP loss. The total uncertainty is a combination of uncertainty from different emission

levels and future socio-economic development ('SSP'), different CMIP6 models with different ENSO amplitude changes ('Climate model'), different choices of discount rate schemes ('Discount rate'), bootstrapped historical response function of global economy to ENSO ('Regression'), and bootstrapped ENSO timeseries under a given amplitude ('ENSO sequence'). All factors are allowed to change for 'Total' uncertainty while only listed factor is allowed to change with the others fixed: 'SSP' fixed to SSP5-8.5, 'Climate model' fixed to the multi-model ensemble median, 'Discount rate' fixed at 3%, 'Regression' fixed to the point estimate, and 'ENSO sequence' fixed to the original ENSO sequence (see 'Assessment of uncertainty' in Methods). Each black vertical line is a point estimate (for example, with four SSP scenarios there are 4 estimates shown for 'SSP' uncertainty), with red vertical line indicating the median.

Reduction in emissions effectively cut the additional loss; under the SSP1-2.6 scenario, a strong mitigation pathway for achieving the Paris Agreement target of limiting warming to 1.5–2.0 °C relative to the pre-industrial level, the additional loss reduces by ~50% from that in the high-emission scenario of SSP5-8.5.

Importantly, there is a nonlinear relationship between changes in ENSO SST variability and additional economic losses, using all CMIP6 models under four IPCC emission scenarios. Models that generate a greater increase in ENSO amplitude tend to generate a greater additional loss in global economic production, and the tendency is systematic and statistically significant (Fig. 4e). This nonlinearity means a heightened risk of increased ENSO to global economy, with many models generating an additional loss in the order of hundreds of trillion US dollars and possibly as high as US$374 trillion, at a 3% discount rate in the high-emission scenario.

There are multiple sources of uncertainty, including model differences, emission scenarios, discount rates used, our econometric model, and ENSO event sequences. We assess the uncertainty by firstly quantifying the relative contribution from these individual factors, one by one by keeping other factors fixed at a chosen level, and subsequently the combined uncertainty (Fig. 4f) (see 'Assessment of uncertainty' in Methods). Uncertainty in climate models, that is, the inter-model differences in ENSO amplitude change, is the largest source of all other factors. Econometric model regression and ENSO sequences contribute to a relatively small uncertainty range, suggesting that our econometric model is reasonably stable and insensitive to economic parameters or random elements such as chaotic processes in the occurrence sequences of ENSO events. That there is a weak sensitivity to ENSO sequences is in contrast to finding of a simultaneous and independent study, which uses a linear econometric model and suggests a large uncertainty source from ENSO event sequences by assuming that El Niño and La Niña exert a perfectly symmetric but opposite impact[47].

Our result of an increased loss in global economy from future ENSO cycles is underpinned by a greater impact of El Niño than that of La Niña and by an increased ENSO variability in the future climate. The greater El Niño impact than that of La Niña is in turn a consequence of the 3-years-long effect on economy that accelerates after the initial shock in the occurrence year; both the initial shock and the lagged effects are far smaller during La Niña. Therefore, a period of ENSO cycles sees a net reduction in global economic production dominated by the loss from extreme El Niño events. Under greenhouse warming, ENSO amplitude increases substantially in all likely emission scenarios, translating to an additional median loss in global economic production in the range of US$14-33 trillion over the last 80 years of the 21st century, that is generally larger in higher emission scenarios; the possibility of an additional loss in hundreds of trillion US dollars cannot be excluded. Achieving the Paris Agreement reduces about half of the increased economic loss. There are transmission pathways of impact from increased future ENSO variability that are not incorporated here, for example, through affecting ocean warming, ice shelf and ice sheet melt, which contribute to sea level rise[48–50] leading to additional economic loss. The additional economic damage from changes in ENSO amplitude in the 21st century highlights the urgency of mitigating emissions of greenhouse gases.

## Methods
### Climate and economic data
To construct ENSO timeseries and its global teleconnection, we use multiple observational and reanalysis datasets: Monthly sea surface temperature data are from Hadley Centre Sea Ice and Sea Surface Temperature dataset (HadISST) (ref. 51), with a 1° grid resolution covering the period of 1950–2021. Monthly land-based surface air temperature and precipitation data are from ECMWF Reanalysis version 5 (ERA5) (ref. 52), with a 0.1° grid resolution for all land areas

covering the period of 1950-2021. These variables are bilinearly interpolated to a horizontal grid of 1° × 1°. Monthly anomalies referenced to the period of 1950-2021 are constructed by removing the monthly climatology, and then quadratically detrended at each grid to remove the long-term greenhouse warming and any low-frequency climate variability. We also use other products such as NOAA Extended Reconstructed SST version 5 (ERSSTv5) (ref. 53) and University of Delaware[54] to confirm robustness of our results.

We use country-specific annual economic data from re-estimating the World Bank Development Indicators[55], which contains Gross Domestic Product (GDP) per capita (in constant 2015 US$, inflation-adjusted) for all countries in the world covering the period of 1960–2019, although data for only a subset of years are available for some countries. We also use the data from Penn World Tables version 10.0 (ref. 56) to test robustness of our results. The population density data is obtained from the Gridded Population of the World (GPW) version 4 (ref. 57).

To project the future change of ENSO under greenhouse warming, we use climate outputs from the state-of-the-art climate models in the Coupled Model Intercomparison Project phase 6 (CMIP6) (ref. 45). The CMIP6 models are forced with historical anthropogenic and natural forcing from 1850 to 2014, and different emission scenarios thereafter from 2015 to 2100. There are four IPCC emission scenarios used, which are SSP1-2.6, SSP2-4.5, SSP3-7.0 and SSP5-8.5, from the scenario for achieving target of Paris Agreement to high-emission scenario. For a given model, outputs might not be available for all emission scenarios, and only the first available experiment from each model for each scenario is used (Supplementary Table S4). All the outputs are bilinearly interpolated to a horizontal grid of 1° × 1°. Monthly anomalies referenced to the period of 1900–1999 are constructed by removing the monthly climatology, and then quadratically detrended at each grid over the 1900–2099 period to remove the long-term trend.

To estimate the economic production loss from future ENSO changes, we use projections from Shared Socioeconomic Pathways database version 2.0 (ref. 46), which consist of several scenarios of projected socio-economic development associated with various degrees of climate forcing over the 21st century. We use the timeseries of country-level economic production (in constant 2015 US$, inflation-adjusted) and population over the period of 2020–2099 under four SSP scenarios (SSP1, SSP2, SSP3, and SSP5) as the baseline to construct counterfactual global GDP growth and then calculated economic loss from ENSO in the future.

### ENSO index and country-specific teleconnection
We use Niño3.4 index to define the ENSO timeseries in the observation and CMIP6 models, which is the monthly SST anomaly averaged in the region of 5°S-5°N, 120°-170°W. Because ENSO generally peaks in boreal winter (December-February, DJF), we use DJF-averaged Niño3.4 index to measure interannual variability of ENSO.

As the economic impact of ENSO is underpinned by the direct climate response to ENSO for individual countries, we construct country-specific ENSO teleconnection to evaluate the extent to which climate of individual countries is affected by ENSO as follows:

(1) We regress normalized monthly grid-point surface air temperature and precipitation anomalies onto normalized D(0)JF(1) Niño3.4 index from May(0) to April(1) (noting the spring barrier[58]), yielding two fields of monthly regression coefficients $\tau_{x,y,m}$ and $\rho_{x,y,m}$ for surface air temperature $T_{x,y,m}$ and precipitation $P_{x,y,m}$ at each longitude-latitude grid point $(x,y)$ and calendar month $m$, respectively. We use partial regression for temperature with precipitation's impact removal to control the correlation between temperature and precipitation, and vice versa.

(2) We accumulate these monthly coefficients from May(0) to April(1) that are statistically significant above the 95% confidence level, and then take the absolute value of the sum to obtain the

cumulative teleconnection strength $\tau_{x,y}$ and $\rho_{x,y}$ (Supplementary Fig. S5).

(3)  We average the grid-point teleconnection strength to country-specific $\tau_i$ and $\rho_i$, weighted by population density in 2020. We use population density weighting rather than other approaches (for example, areal weighting) for its better representation of reaction of human-based economic activity to climate anomaly within a given country. Economic impact transmitted from climate shock is generally driven by human interactions and transactions. Population density weighting puts more emphasis on areas with higher concentration of people, which are typically the centers of economic activity. However, as we have shown, a common shock to most countries dominates, rendering our results insensitive to approaches adopted.

(4)  We accumulate the country-specific temperature and precipitation teleconnection strengths as the total cumulative teleconnection strength $\psi_i$ (Supplementary Fig. S6). We assign those countries with $\psi_i > 0.5$ to the "teleconnected" group, and $0 < \psi_i \leq 0.5$ to the "weakly-teleconnected" group (Fig. 1a; Supplementary Fig. S6b).

## Empirical econometric model

A distributed-lag timeseries model estimated by the Ordinary Least Squares (OLS) is applied to assess the nonlinear effect of ENSO on global economic production. Economic growth for each country is represented by the first difference of natural logarithm of annual GDP per capita. Combining the factors including observed climate variables and unobserved time-invariant and time-trending covariates, we then build the model as:

$$\Delta \log(y_{it}) = \sum_{l=0}^{n} \{ \alpha_{1,l} E_{t-l} + \alpha_{2,l} E_{t-l}^2 + \beta_{1,l} T_{it-l} + \beta_{2,l} T_{it-l}^2 \\ + \lambda_{1,l} P_{it-l} + \lambda_{2,l} P_{it-l}^2 \} + \mu_i + \theta_{1i} t + \theta_{2i} t^2 + \varepsilon_{it} \quad (1)$$

where $y_{it}$ is GDP per capita in country $i$ and year $t$, $l$ is the lag year to year $t$, $E$ is DJF-averaged Niño3.4 index in year $t - l$, $T$ and $P$ are annual mean surface air temperature and precipitation in year $t - l$, after removing the ENSO's signal through linear regression. Consistent with previous literatures with respect to ENSO's socio-economic impact[20,28], we include the country-fixed effect ($\mu_i$) as a "control" variable that represents time-invariant factors such as history, culture backgrounds and geographic location, and the country-specific linear and quadratic time trend ($\theta_{1i} t + \theta_{2i} t^2$) as "control" factors that change over time within a country, such as development, trade liberalization and technological progress. We also incorporate specific control variables such as trade openness, share of agricultural GDP and financial depth as sensitivity tests, and our estimation is insensitive to these factors. We drop the year-fixed effect that represents time-varying factors common across countries such as global recession, since the year-fixed effect introduces high risk of collinearity as ENSO timeseries could be correlated with time-specific factors, making it harder to disentangle the separate impacts from ENSO and time-fixed effects. ENSO can be treated as a time-specific global phenomenon that commonly affects countries. Including year-fixed effects weakens the statistical influence of ENSO, leading to an underestimation of the real impact of ENSO on economic growth[59]. In contrast to previous studies which depict different impact of El Niño and La Niña through a Heaviside step function[22,23], we incorporate a continuous nonlinear effect of ENSO in a quadratic function in which both the linear and the nonlinear components include lagged effects. These lag terms account for growth effects after the contemporaneous climate shock of an ENSO event, allowing subsequent acceleration in the impact on economic activity after the initial shock.

Equation (1) is simultaneously fitted by all the country-year samples from 181 countries over the period of 1961–2019 ($N = 7404$). Inclusions of annual mean temperature and precipitation are aimed to parallelly estimate the impact from mean state change, with assumption that ENSO-induced temperature and precipitation anomalies are independent to the mean change. We test this assumption by removing ENSO signal from annual mean temperature and precipitation by linear regression, which goes

$$T_{it}^* = T_{it} - E_t \cdot r_i(T_i, E) \quad (2)$$

where $T_{it}^*$ is the annual temperature after ENSO signal removal in country $i$ and year $t$. $r_i(T_i, E)$ is the linear regression coefficient of annual temperature $T_i$ onto the DJF Niño3.4 index $E$ in country $i$ during the period of 1950-2021. Same process is applied for annual mean precipitation $P_{it}$. There is little difference in the coefficients of ENSO impact after removing ENSO signal from annual mean temperature and precipitation.

We test different lag years to obtain the optimum one for ENSO's growth effect. ENSO continue to have negative effect till year 3, after which time little further impact is seen and uncertainty increases (Supplementary Fig. S2; Supplementary Table S1). As such, a 3-year lag is applied ($n = 3$) to estimate the growth effect of ENSO. We estimate loss from ENSO in each year as:

$$\Delta g_t^E = \exp \left\{ \sum_{l=0}^{n} (\alpha_{1,l} E_{t-l} + \alpha_{2,l} E_{t-l}^2) \right\} - 1 \quad (3)$$

where $\Delta g_t^E$ is the ENSO-induced change in growth rate of annual GDP per capita in year $t$. Hence, the in-dollar-value impact of ENSO in year $t$ is obtained by

$$\Delta Y_t^E = \Delta g_t^E \cdot \sum_i y_{it} \quad (4)$$

where $\Delta Y_t^E$ is the ENSO-induced global per capita GDP change in year $t$.

## Counterfactual ENSO and scenario

To avoid any influence from sequence of ENSO events when assessing change of ENSO-induced economic growth reduction under greenhouse warming, we develop counterfactual scenarios with hypothetical future ENSO timeseries. The counterfactual future ENSO is re-scaled from original timeseries in the 21st century to be of the same amplitude as in the 20th century, that is:

$$E^C = E^{SSP} \cdot \frac{\sigma(E^{HIST})}{\sigma(E^{SSP})} \quad (5)$$

where $E^C$ is the counterfactual Niño3.4 timeseries, $E^{HIST}$ and $E^{SSP}$ are Niño3.4 timeseries under the 20th and 21st century, respectively. This approach retains the temporal evolution of ENSO in a SSP scenario to remove any uncertainty that might arise from the sequence of individual El Niño and La Niña events (Supplementary Fig. S10a). The growth rate difference between the original and counterfactual scenarios in each year $\Delta g_t^E$ is calculated as:

$$\Delta g_t^E = \exp \left\{ \sum_{l=0}^{n} (\alpha_{1,l} E_{t-l}^{SSP} + \alpha_{2,l} (E_{t-l}^{SSP})^2) \right\} - \exp \left\{ \sum_{l=0}^{n} (\alpha_{1,l} E_{t-l}^C + \alpha_{2,l} (E_{t-l}^C)^2) \right\} \quad (6)$$

To project the economic development in global GDP absent ENSO in the future, we build a counterfactual global GDP growth based on SSP projections. We assume that ENSO permanently reshapes the global economy as its impact in each year aggregated over a long-term period. The counterfactual GDP growth starts at year 2020, in which the GDP is the same as particular SSP pathways. Then the cumulative

growth rate reduction from ENSO in each year (both contemporaneous effect at that year and growth effect from previous years) is restored to the original GDP growth rate. As such, the counterfactual GDP is generated and aggregated to the end of 21st century, which is calculated as:

$$y_t^C = y_{t-1}^C \cdot (1 + g_t + \Delta g_t^E) \qquad (7)$$

where $y_t^C$ is the counterfactual global GDP per capita at year $t$, $g_t = \frac{y_t}{y_{t-1}} - 1$ is the growth rate at year $t$ based on the original output of SSP database (Supplementary Fig. S10b).

## Statistical significance test

We implement several sets of Bootstrap method[60] to assess statistical significance of the historical response function and multi-model projections. To quantify uncertainty in estimates of $\alpha_1$ and $\alpha_2$, we apply different bootstrap strategies (1) Sampling by country in which the 181 countries are randomly resampled to construct another 10,000 realizations of 181-element lists. In the resampling process, any country is allowed to be selected again. These resampled lists of countries are used to re-estimate the Eq. (1) and obtain the estimates of $\alpha_1$ and $\alpha_2$. (2) Sampling by year, in which the 59 years from 1961–2019 period are randomly resampled to construct another 10,000 realizations of 59-element lists. In the resampling process, any year is allowed to be selected again. These resampled lists of years are used to re-estimate the Eq. (1) and obtain the estimates of $\alpha_1$ and $\alpha_2$. (3) Sampling by 5-year block, in which we divide the country-year data into 5-year blocks (like 1961–1965, 1966–1970 and so on), and resample these blocks as (2). The 1.96 s.d. of the 10,000 estimates of $\alpha_1$ and $\alpha_2$ represents the 95% confidence interval. We use strategy (1) in our presentation of results, but we also show that the robustness of our historical response function is insensitive to alternative bootstrap strategies (Supplementary Fig. S3).

We also apply several sensitivity tests by omitting some of data and re-estimating Eq. (1). In one such test, we re-estimate the econometric model under 50-year running window periods of 1960–2009, 1961–2010, …, and 1970–2019. In another, we randomly drop 3 individual years of data from the period of 1960–2019 to re-estimate the econometric model for 1000 times by the Bootstrap method. We find that our historical response function is insensitive to these tests (Supplementary Fig. S4).

A similar Bootstrap method allowing repeat is used to test statistical significance of a difference between the 20th and 21st century, or between a counterfactual and its original SSP scenario in the 21st century, in multi-model ensemble mean Niño3.4 variability and economic growth reduction. Specifically, their individual values (corresponding to models) in each century are resampled randomly to construct 10,000 realizations of multi-model ensemble mean values, and standard deviation of the 10,000 realizations for each century is calculated. When the multi-model ensemble mean difference is greater than the sum of s.d. values of the 10,000 realizations of the two centuries, the multi-model ensemble mean difference is statistically significant above the 95% confidence level (Supplementary Fig. S9).

## Heterogeneity of response function

To test the heterogeneity of ENSO effect across different groups of countries (Fig. 1), we allow interaction between both the linear and quadratic terms of Niño3.4 and an indicator $D_i$ for whether a country is, for example, a teleconnected country. We then incorporate it into the econometric model by

$$\Delta \log(y_{it}) = \sum_{l=0}^{n} \{ \alpha_{1,l} E_{t-l} + \alpha_{2,l} E_{t-l}^2 + D_i(\alpha_{3,l} E_{t-l} + \alpha_{4,l} E_{t-l}^2) + \beta_{1,l} T_{it-l} \\ + \beta_{2,l} T_{it-l}^2 + \lambda_{1,l} P_{it-l} + \lambda_{2,l} P_{it-l}^2 \} + \mu_i + \theta_{1i} t + \theta_{1i} t^2 + \varepsilon_{it} \qquad (8)$$

$$D_i = \begin{cases} 1 & \text{if country } i \text{ is a teleconnected country} \\ 0 & \text{if country } i \text{ is a weakly} - \text{teleconnected country} \end{cases}$$

where $\alpha_{1,l}$ and $\alpha_{2,l}$ represent the linear and nonlinear coefficients of the response function for weakly-teleconnected countries, and $\alpha_{3,l}$ and $\alpha_{4,l}$ represent the adjustments to $\alpha_{1,l}$ and $\alpha_{2,l}$ that are only applicable to teleconnected countries. Thus, the response functions of teleconnected and weakly-teleconnected countries are statistically different if both $\alpha_{3,l}$ and $\alpha_{4,l}$ are significantly distinguishable from zero ($p < 0.05$). We apply the same approach to agriculture-dependent/-independent and high-income/lower-income countries. We find that while some coefficients of linear terms $\alpha_{3,l}$ are significant, coefficients of nonlinear terms $\alpha_{4,l}$ are indistinguishable from zero for these groups of countries (Supplementary Table S2). This means that the common nonlinear nature reflects the feature that the global spillovers and cascading effects dominate global economic impact of ENSO. On the other hand, the heterogeneous magnitude of response is due to several different factors. One is linked to geographical locations in that strongly-teleconnected countries tend to be in the tropics and lower-income countries; another is economic structure in that there tends to be a high proportion of agriculture in GDP in lower-income countries; further, in lower-income countries, the level of preparedness and the capacity to mitigate are relatively low.

To test any country-level heterogeneity of ENSO effect across individual countries, we include the interaction of the common ENSO shock with country-specific teleconnection in the econometric model by

$$\Delta \log(y_{it}) = \sum_{l=0}^{n} \{ \alpha_{1,l} E_{t-l} + \alpha_{2,l} E_{t-l}^2 + \gamma_{1,l} \psi_i E_{t-l} + \gamma_{2,l} (\psi_i E_{t-l})^2 + \beta_{1,l} T_{it-l} \\ + \beta_{2,l} T_{it-l}^2 + \lambda_{1,l} P_{it-l} + \lambda_{2,l} P_{it-l}^2 \} + \mu_i + \theta_{1i} t + \theta_{2i} t^2 + \varepsilon_{it} \qquad (9)$$

where $\psi_i E_{t-l}$ is the interaction term between the common ENSO shock and country-specific teleconnection. We find that coefficients of the interaction terms only show a weak statistical significance (Supplementary Table S3), suggesting a major contribution of common ENSO shock to the global impact.

## Assessment of uncertainty

Our projection of additional loss in global economic production contains several sources of uncertainty, including various future socio-economic baseline and climate forcing from different SSP scenarios, varying changes in ENSO amplitude across climate models, choices of discount rate, and historical function regression. Given the counterfactual GDP timeseries is associated with ENSO timeseries, the sequence of ENSO is also a potential factor of uncertainty. We therefore re-arrange the ENSO timeseries in each CMIP6 model to construct 10,000 timeseries, which have the same amplitude but different sequences of ENSO events. To quantify the relative contribution from the above factors to uncertainty, we hold four of five factors fixed and allow the fifth to change. The factors are fixed as follows: SSP scenario is fixed to SSP5-8.5, climate model projection of ENSO changes is fixed to the multi-model ensemble median, discount rates are fixed at 3%, historical function regression is fixed to the point estimate, and ENSO sequence is fixed to the original sequence of timeseries as simulated in each climate model. For the total uncertainty, each factor is allowed to vary.

The total uncertainty leads to a 95% confidence interval of changes in global economic loss from -US$247 trillion to US$20 trillion, while SSP scenario uncertainty alone leads to a 95% confidence interval of -US$36 trillion to -US$14 trillion, climate model uncertainty to a 95% confidence interval of -US$290 trillion to US$25 trillion, discount rate

uncertainty to a 95% confidence interval of -US$90 trillion to -US$15 trillion, historical regression uncertainty to a 95% confidence interval of -US$55 trillion to -US$22 trillion, and ENSO sequence uncertainty to a 95% confidence interval of -US$47 trillion to -US$38 trillion (Fig. 4f). Climate models with various ENSO amplitude changes contribute most to the total uncertainty, suggesting the projected additional economic loss is highly dependent on ENSO amplitude change in the future.

## Data availability

All datasets related to this paper are publicly available and can be downloaded from the following websites: HadISSTv1.1: https://www.metoffice.gov.uk/hadobs/hadisst/, ERA5: https://www.ecmwf.int/en/forecasts/dataset/ecmwf-reanalysis-v5, ERSSTv5: https://psl.noaa.gov/data/gridded/data.noaa.ersst.v5.html, University of Delaware: https://psl.noaa.gov/data/gridded/data.UDel_AirT_Precip.html, World Bank Development Indicators: https://databank.worldbank.org/source/world-development-indicators, Penn World Tables v10.0: https://www.rug.nl/ggdc/productivity/pwt/, GPWv4.11: https://sedac.ciesin.columbia.edu/data/set/gpw-v4-population-density-adjusted-to-2015-unwpp-country-totals-rev11, CMIP6: https://esgf-node.llnl.gov/search/cmip6/ SSP database version 2.0: https://tntcat.iiasa.ac.at/SspDb/dsd?Action=htmlpage&page=10.

## Code availability

Codes for the main results are available on Zenodo at https://zenodo.org/record/8238350.

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

## Acknowledgements
This study is supported by the National Natural Science Foundation of China (42176218, Y.Z.) and the Strategic Priority Research Program of the Chinese Academy of Sciences (XDB40030000). X.L. and Z.L. are supported by the National Natural Science Foundation of China (41925025 and 92058203). Y.L. is supported by the Fundamental Research Funds for the Central Universities (202261003) and the China Scholarship Council (202106330019).

## Author contributions
Y.L. and W.C. conceived this study and wrote the initial manuscript. Y.L. performed the analyses in discussion with W.C., X.L., Z.L. and Y.Z. All authors contributed to interpreting results, discussion and improvement of this paper.

## Competing interests
The authors declare no competing interests.
