## [Peer Review File · Nature Communications]

Nonlinear El Niño impacts on the global economy under climate changeREVIEWER COMMENTS

Reviewer #1 (Remarks to the Author):

This is an interesting paper. The role of the ENSO cycles on world economies has been investigated before, but this study presents additional details to the discussion. Specifically, it allows for nonlinear effects of ENSO--basically allowing for the possibility that both El Nino and La Nina events can be damaging to economies.

One of the key messages of the study, which is presented as its main "selling point" is that the estimated effect in this study is the order of magnitude larger than that presented in previous studies. The paper doesn't indicate why that may be the case. Presumably, the authors allude that their estimates are correct and therefore estimates of the previous studies are wrong, and that may very well be the case, but it should be noted, rather explicitly, what are the key points of departure. Is it data-related (because the present study uses more recent data) or the econometric technique? In any case, it also would be interesting to see if the present study replicates the results of the previous studies.

This study seems to be using nominal GDP per capita. If so, then the effects of different historical events are not directly comparable. 100 million dollars in 1998 were worth more than 100 million dollars in 2015, for example.

The dependent variable is the natural log of the GDP per capita. Meaning that the estimated parameters are, approximately, semi-elasticities. But that also puts the high-income countries and low-income countries in the same basket. It seems like the authors subsequently relax this homogeneity assumption, but it is not immediately clear how the estimated impacts of ENSO events, in dollar value, have been obtained.

The effect on teleconnected and not teleconnected (or weakly teleconnected) countries seem to be very similar (e.g., Fig1), which raises the concern of whether this estimated effect is spurious. I must apologize for this comment because I have no reason to make such a claim, other than a hunch. But I decided to voice this concern, just to be honest with myself, at the very least.

It would have been nice to have, at least, the main regression results presented in a table, just to get a better idea of which parameters are driving the results.

Reviewer #2 (Remarks to the Author):

The paper is engaging and well-written. It tackles a relevant issue, which could spur interest in a large community of readers. In particular, the authors study the effects of El Nino and la Nina events on economic growth, using well-established empirical methods from climate econometrics and a global panel of countries over the period 1961-2019. My feeling is that the paper has potential, but the authors should convincingly address some key issues, which I think currently undermine the validity and innovativeness of the study.

First, a suggested interpretation of the estimated effects is grossly missing. The authors estimate economic losses from ENSO by means of an econometric model relating country-specific economic growth rates to an ENSO index, country-specific temperatures, and country-specific precipitation (with multiple powers and lags). They then use the marginal estimated effects to compute losses stemming from ENSO oscillations (please, pay attention to my third point below). However, they are silent on what mechanisms could drive economic losses from El Nino/La Nina. I feel this is relevant, especially when the effect of local weather change is controlled for. Intuitively, one would expect that El Nino

events cause changes in the experienced weather conditions in different countries, which - in turn - affect local economic activity and production. However, given the model estimated by the authors, such effects should be largely captured by the precipitation and temperature variables. Hence, the "additional effect" of ENSO is very interesting and surprisingly large, but - for this to be credible - I feel the authors should make an interpretative effort, suggesting what mechanisms may lie behind the statistical relationship the paper seems to spot robustly. This is even more relevant as the present study reports much larger losses than in previous literature.

Second, the innovativeness of the paper should be streamlined much more effectively. The authors sometimes seem to motivate their study on the basis of inconclusive and scant evidence about the effects of El Nino/La Nina on the macroeconomy (see e.g. the abstract at lines 18-19). While I agree that further studies are definitely necessary, I have some issues grasping the novelty of the present paper with respect to the literature. For example, Smith and Ubilava (2017, GEC) - which the authors cite - offer a quite similar approach, finding non-linear effects, spanning across different years and varying across countries and climatic zones. Cashin et al. 2017 (2017, JIntEco) - which the authors cite - offer a much more comprehensive assessment of the macroeconomic effects of El Nino and La Nina events, though with a different methodology. I feel the authors should make crystal clear the piece of evidence they add to the literature. My reading is that the originality lies in i. the role of teleconnections (which would deserve more space in the paper), ii. the additional effect of ENSO beyond that of local temperature and precipitations (see my point above), iii. the projected effects on growth, which are very large and need to be robust (see my third and fourth points immediately below).

Third, the authors estimate a growth effect while they mostly report results in terms of a level effect. While the paper finds a robust non-linear effect of ENSO on growth, economic losses are mostly reported in terms of trillions of dollars of economic damage, which is suggestive of a level effect. I feel this approach is problematic for two reasons: first, it does not account for the debate on the persistence of climate impacts on the macroeconomy (see e.g. Diaz and Moore 2017, NCC; Bastien-Olvera et al. 2021, ERL); second, it makes results more difficult to interpret from a quantitative standpoint. Indeed, 1 trillion USD is relatively low for the US economy, while extremely large for a small African country; similarly, it is difficult to say whether a loss of 2.1 trillion (a figure found in the abstract) is large or small for the global economy without a meaningful term of comparison (e.g. the value of the global GDP in the year of the shock or the loss generated by another event, e.g. the global financial crisis). Further, while the authors account for inflation, it is not always clear in dollars of which year results are computed. More relevantly, I failed to get the exact meaning of equations 2 and 4. To my reading, they do not reflect the impact on growth the authors would like to measure. I would kindly encourage the authors to provide more details, possibly together with examples for some countries. Indeed, it is not evident what losses in which countries are driving the global loss estimates the authors come up with. More clarity along these lines would make the results much more trustworthy. Unpacking the results across countries would offer an intriguing perspective, but this is just a suggestion.

Fourth. In their projection exercises, it is unclear what drivers of growth the authors consider and what they do not. For example, in the seminal paper by Burke et al. (2015, Nature) projections include socio-economic trends (proxied by historical growth rates of GDP). I have the impression the authors exclude such trends when projecting the GDP of various countries. For example, this seems to be the case in the section "Substantial additional loss in economic production from increased ENSO amplitude" (from line 197). Accounting for such trends may change the projection results sensibly. Similarly, accounting for future temperature will affect growth rates; however, the counterfactual scenario seems to assume that ENSO will change alongside future climatic and SSP scenarios while temperatures are not. I think the authors should either change their projections or offer a crystal-clear and convincing explanation of their counterfactual and projection exercises.

Fifth, aggregation with discount rates may be problematic. Impacts on economic production are

aggregated with discount rates chosen by the authors. Figure 4 is the most communicative example. I find this choice debatable. First, the last two decades have witnessed a fierce debate about the selection of discount rates, and results from impact studies typically showed large sensitivity to such a parameter. This paper is not an exception. Second, it is not clear to me why the authors need to discount back in time a projected impact. For example, they could just compute the loss as the ratio between the projected global GDP accounting for ENSO effects and the projected GDP without the ENSO effect. This would be more similar, for example, to the approach by Burke et al. 2015 and to the majority of damage functions.

Finally, let me provide few other yet less substantive comments.

- at lines 95-98 the authors state they include mean temperature and precipitation in their model to estimate the effects of mean state change and introduce the assumption that ENSO-induced temperature and precipitation changes are independent of mean state changes. First, I am not convinced that equation 1 reflects this assumption (rather, it reflects yearly deviation from country-specific trends that may be caused by any events, including ENSO); second, I am not convinced the assumption of independence is realistic and/or innocent.
- the effects of temperature and precipitations are not discussed and it is also not clear whether these variables are included in the estimation in their population-weighted version or not. This may have implications for how the authors capture exposure of economic production to the weather and for the interpretation of results.
- The links with the blossoming climate econometrics literature could be better surveyed and the paper better positioned in such a stream of studies. For example, El Nino and La Nina are known to exert impacts through changes in precipitation patterns (among other channels). Palagi et al. (2022, PNAS) and Kotz et al. (2022, Nature) are two extremely recent contributions emphasizing the impact of precipitation anomalies on growth and the macroeconomy. The authors could leverage these results to provide a more informative contextualization of their findings. This links back to my first point.

References

- Bastien-Olvera, B., Granella, F. & Moore, F. (2021). Persistent effect of temperature on GDP identified from lower frequency temperature variability, *Environmental Research Letters*, 17 (8).
- Burke, M., Hsiang, S. M., & Miguel, E. (2015). Global non-linear effect of temperature on economic production. *Nature*, 527(7577), 235-239.
- Cashin, P., Mohaddes, K., & Raissi, M. (2017). Fair weather or foul? The macroeconomic effects of El Niño. *Journal of International Economics*, 106, 37-54.
- Diaz, D., & Moore, F. (2017). Quantifying the economic risks of climate change. *Nature Climate Change*, 7(11), 774-782.
- Kotz, M., Levermann, A., & Wenz, L. (2022). The effect of rainfall changes on economic production. *Nature*, 601(7892), 223-227.
- Palagi, E., Coronese, M., Lamperti, F., & Roventini, A. (2022). Climate change and the nonlinear impact of precipitation anomalies on income inequality. *Proceedings of the National Academy of Sciences*, 119(43), e2203595119.
- Smith, S. C., & Ubilava, D. (2017). The El Niño Southern Oscillation and economic growth in the developing world. *Global Environmental Change*, 45, 151-164.

Reviewer #3 (Remarks to the Author):

This article, which aims to assess the overall economic cost of ENSO in a context of climate change, covers an important subject area. My comments focus on how the ENSO impact is considered and the relevance of an average assessment of its cost in terms of overall GDP (knowing that I am not a climatologist). I find many issues that must be addressed for the assessment to be credible.

My comments relate to the estimation of the country-specific teleconnection strengths as well as the estimation of the economic cost of ENSO-related events. It seems fundamental to me to solidify these results before any climate projection.

1. ENSO index and country-specific teleconnection

The authors propose a measure of the degree of teleconnection specific to each country. More detail should be given regarding the construction of the indicator reflecting the magnitude of teleconnection patterns.

The estimation of the regression coefficients ρ and τ raises questions in the absence of a clearly specified model. Are the temperature and precipitation anomalies, used as explained variables, standardized? Are precipitation included as a covariate in the regression when analyzing temperature and vice versa to control for the correlation between temperature and precipitation.

Moreover, several factors can lead to a waxing and waning in the strength of the ENSO teleconnections: changes in the evolution of the SST pattern during ENSO, changes in the ENSO's temporal behavior, changes in the mean base climatology, and its modulation of the teleconnection process, and simple sampling variations. It could then be helpful to review the robustness of the country-specific teleconnection metric assessed by the authors by analyzing its temporal stationarity throughout the entire reference period.

In line 403, the authors state that the estimated coefficients are weighted by population density in 2020. First, what is the source of the population density data? Second, the authors should give better justification with regard to this choice concerning the spatial aggregation procedure. Number of economic studies aggregate ENSO teleconnections within countries borders using a fixed set of population weights, so that the relevant concept is the average weather experienced by a person in the administrative area, not the average weather experienced by a place. However, this procedure implies that the aggregation gives a higher weight to highly populated (urban) areas while areas characterized by low population densities may play a major economic role and while being impacted by ENSO shocks. This is particularly the case for agricultural regions, a sector through which the majority of ENSO-related economic shocks spread.

In line 405, the authors divide the countries into two categories based on the accumulation of the country-specific temperature and precipitation teleconnection strengths (ψ_i) so that countries where $\psi_i < 0.5$ are weakly connected and countries where $\psi_i > 0.5$ are connected. This categorization is of particular concern to me. On what basis is this classification made? It seems from Figure S4 that the choice of such a low threshold value completely crushes the country heterogeneity of ENSO shocks and that almost all countries belong to the category of teleconnected countries. In other words, what is the point of calculating a specific indicator for each country to ultimately arrive at such a broad categorization? It seems to me, moreover, that this methodological choice is decisive for the rest of the article and therefore the determination of the overall cost of ENSO.

It would be necessary to consider alternative global measurements of the level of teleconnection based on more precise measurements (spatial variability, spatial extent, etc.) and to accompany the results with a map showing the level of teleconnection by grid point (before the aggregation at the country level).

2. Empirical econometric model

The authors test a distributed-lag timeseries model to estimate the economic cost of ENSO events. Is this model estimated by the GLS or by the OLS? I think it is essential to have more information on the possible presence of multicollinearity (using the VIF for example). Indeed, the measurement of ENSO shocks being indexed only by time (shock common to all countries), the risk of quasi collinearity with the country fixed effects (μ_i) is high and likely to bias the estimates.

Using a common ENSO shock to estimate its overall cost does not seem to constitute a significant contribution compared to the previous literature on the subject which explicitly introduces a measure of the strength of the teleconnection specific to each country in the econometric specification or via the interaction between the common shock and country specific weather anomalies. Why not integrate the results from the calculation of country-specific teleconnections within the econometric specification? Indeed, the distinction made in Figure 1 does not seem to show any difference between the categories of countries and seems insufficient.

The distinction between rich and poor countries should also be clarified. On what basis is this distinction made? This distinction may seem tautological from an economic point of view, as poor countries are more vulnerable to climatic hazards... A more relevant distinction should be made on the basis of the countries structural characteristics by distinguishing, for example, countries dependent on agriculture, countries whose economic fabric is more diversified and thus evaluating the heterogeneity estimated cost of ENSO events.

The annual measurement of the ENSO indicator (Nino 3.4) should be subject to robustness. The authors should consider comparing their non-linear (quadratic) measurement to a measurement based on event occurrence (based on threshold values) thus justifying the methodological contribution of the paper with respect to the prior literature relying on a Heaviside function in order to dissociate El Niño events from La Nina events. It is important to note that the references mentioned by the authors use both a Heaviside function and a quadratic specification in order to characterize ENSO events within the framework of their econometric specifications.

Although the measure of the climate can reasonably be considered exogenous (not invalidating the hypotheses of the model), it would seem relevant to include control variables reflecting the characteristics of the countries such as the share of GDP in the Value added Agricultural, the degree of trade openness.

Response to Reviewer #1

This is an interesting paper. The role of the ENSO cycles on world economies has been investigated before, but this study presents additional details to the discussion. Specifically, it allows for nonlinear effects of ENSO--basically allowing for the possibility that both El Nino and La Nina events can be damaging to economies.

We thank the reviewer for the positive comments.

One of the key messages of the study, which is presented as its main "selling point" is that the estimated effect in this study is the order of magnitude larger than that presented in previous studies. The paper doesn't indicate why that may be the case. Presumably, the authors allude that their estimates are correct and therefore estimates of the previous studies are wrong, and that may very well be the case, but it should be noted, rather explicitly, what are the key points of departure. Is it data-related (because the present study uses more recent data) or the econometric technique? In any case, it also would be interesting to see if the present study replicates the results of the previous studies.

Previous estimates included direct and contemporaneous losses from El Niño/La Niña events, and produced results that are consistent with ours. For example, Smith and Ubilava (2017) and Generoso et al. (2020) used a Heaviside step function and allowed limited lags to describe the different impact between El Niño and La Niña. Their approach therefore assumed the same impact sensitivity for weak and strong events. Although the resultant loss is smaller, it is of the same order of magnitude.

Here we take into account, in our econometric model, the likelihood of an impact increasing or changing nonlinearly with amplitude of El Niño/La Niña, and include all possible lags. Because of these two improvements, our continuously nonlinear econometric model (**Fig. R1A**), while capturing main features of previous models, generates a greater loss than that in previous studies. The continuously nonlinear model is important for assessing future ENSO impact as ENSO amplitude is projected to increase under greenhouse warming.

We have discussed this point in Line 82-86.

Fig. R1A. Comparison of (a) results from the fully nonlinear model in our paper (same as Fig. 1a) and (b) the Heaviside model, both with contemporaneous effect and 3-year lagged effect included; also the comparison of cumulative losses from observed extreme El Niño events from (a) the fully nonlinear model in our paper (same as Fig. 2a) and (b) the Heaviside model. Comparison of (c) cumulative losses from observed extreme El Niño events under the fully nonlinear model in our paper (same as Fig. 2a) and (d) the Heaviside model.

This study seems to be using nominal GDP per capita. If so, then the effects of different historical events are not directly comparable. 100 million dollars in 1998 were worth more than 100 million dollars in 2015, for example.

We use constant 2015 US\$ GDP (Line 408 and 429). We have made this clearer.

The dependent variable is the natural log of the GDP per capita. Meaning that the estimated parameters are, approximately, semi-elasticities. But that also puts the high-income countries and low-income countries in the same basket. It seems like the authors subsequently relax this homogeneity assumption, but it is not immediately clear how the estimated impacts of ENSO events, in dollar value, have been obtained.

We have improved this in the revised version. We have now added more detailed steps to Eq. 2 to explain how the ENSO-induced change in GDP growth rate is calculated. Given that

$$\Delta \log(y_{it}) = \log(y_{it}) - \log(y_{it-1}) = \log\left(\frac{y_{it}}{y_{it-1}}\right) = \log(1 + g_{it})$$

Then

$$\Delta g_t^E = \exp\left\{\sum_{l=0}^n (\alpha_{1,l} E_{t-l} + \alpha_{2,l} E_{t-l}^2)\right\} - 1$$

where Δg_t^E is the ENSO-induced change in growth rate of annual GDP per capita in year t .

Hence the in-dollar-value impact of ENSO in year t is obtained by

$$\Delta Y_t^E = \Delta g_t^E \cdot \sum_i y_{it}$$

where ΔY_t^E is the ENSO-induced global per capita GDP change in year t .

The effect on teleconnected and not teleconnected (or weakly teleconnected) countries seem to be very similar (e.g., Fig1), which raises the concern of whether this estimated effect is spurious. I must apologize for this comment because I have no reason to make such a claim, other than a hunch. But I decided to voice this concern, just to be honest with myself, at the very least.

We thank the reviewer for raising this point, which lets us revisit the heterogeneity and homogeneity of ENSO impact.

First, we have revised **Fig. 1** by changing the threshold of teleconnected countries to $\psi > 1$, and then depicting the comparison between teleconnected and weakly-teleconnected countries (**Fig. 1b**), agriculture-dependent and -independent countries (**Fig. 1c**), rich and poor countries (**Fig. 1d**), to show more relevant distinctions.

Next, we have examined whether the response functions across different groups of countries are statistically distinguishable. We allow interaction between both the linear and quadratic terms of Niño3.4 and an indicator D_i for whether a country is, for example, a teleconnected country. We then incorporate it into the econometric model (**Eq. 1**):

$$\Delta \log(y_{it}) = \sum_{l=0}^n \left\{ \alpha_{1,l} E_{t-l} + \alpha_{2,l} E_{t-l}^2 + D_i (\alpha_{3,l} E_{t-l} + \alpha_{4,l} E_{t-l}^2) + \beta_{1,l} T_{t-l} + \beta_{2,l} T_{t-l}^2 + \lambda_{1,l} P_{t-l} + \lambda_{2,l} P_{t-l}^2 \right\} + \mu_i + \theta_{it} + \theta_{it}^2 + \varepsilon_{it}$$

$$D_i = \begin{cases} 1 & \text{if country } i \text{ is a teleconnected country} \\ 0 & \text{if country } i \text{ is a weakly-teleconnected country} \end{cases}$$

where $\alpha_{1,l}$ and $\alpha_{2,l}$ represent the linear and nonlinear coefficients of the response function for weakly-teleconnected countries, and $\alpha_{3,l}$ and $\alpha_{4,l}$ represent the adjustments to $\alpha_{1,l}$ and $\alpha_{2,l}$ that are only applicable to teleconnected countries. Thus, the response functions of teleconnected and weakly-teleconnected countries are statistically different in structure if both $\alpha_{3,l}$ and $\alpha_{4,l}$ are significantly distinguishable from zero ($p < 0.05$). We apply the same approach to agriculture-dependent/-independent and rich/poor countries. We find that while some coefficients of linear terms $\alpha_{3,l}$ are significant, coefficients of nonlinear terms $\alpha_{4,l}$ are indistinguishable from zero (**Table R1A**).

Therefore, we conclude that although the heterogeneity of ENSO impact does exist, for example, teleconnected countries exhibit different asymmetry of the ENSO impact, with somewhat greater negative effect from El Niño than weakly-teleconnected countries, the

overall response functions are statistically indistinguishable between each other, indicating a dominant homogeneity of the nonlinear effect for all groups of countries.

Table R1A. Same as Table S1, but with regression coefficients of interaction terms (α_3, α_4) added.

	Teleconnected/Weakly-teleconnected countries	Agriculture-dependent/-independent countries	Rich/Poor countries
$\alpha_{1,0}$	-0.0055***	-0.0053***	-0.0053***
$\alpha_{1,1}$	-1.53×10^{-4}	-0.0023**	-0.0029***
$\alpha_{1,2}$	0.0011	1.64×10^{-4}	-0.0017*
$\alpha_{1,3}$	5.80×10^{-4}	4.74×10^{-4}	-0.0012
$\alpha_{2,0}$	7.67×10^{-4}	5.88×10^{-4}	-1.31×10^{-5}
$\alpha_{2,1}$	-0.0029***	-0.0025***	-0.0024***
$\alpha_{2,2}$	-0.0033***	-0.0029***	-0.0030***
$\alpha_{2,3}$	-4.41×10^{-4}	-8.07×10^{-4}	-0.0017**
$\alpha_{3,0}$	5.00×10^{-4}	5.31×10^{-4}	4.37×10^{-4}
$\alpha_{3,1}$	-0.0046***	-9.81×10^{-4}	6.87×10^{-4}
$\alpha_{3,2}$	-0.0033**	-0.0022	0.0026*
$\alpha_{3,3}$	-0.0029**	-0.0040***	4.86×10^{-4}
$\alpha_{4,0}$	-8.78×10^{-4}	-8.01×10^{-4}	7.26×10^{-4}
$\alpha_{4,1}$	0.0013	0.0011	7.98×10^{-4}
$\alpha_{4,2}$	0.0011	7.80×10^{-4}	8.24×10^{-4}
$\alpha_{4,3}$	-0.0013	-9.35×10^{-4}	0.0013
R^2	0.2064	0.2056	0.2047

It would have been nice to have, at least, the main regression results presented in a table, just to get a better idea of which parameters are driving the results.

We included all the regression coefficients with their statistical significance in **Table S1** of Supplementary Information.

Thank you again for your helpful and encouraging comments.

Response to Reviewer #2

The paper is engaging and well-written. It tackles a relevant issue, which could spur interest in a large community of readers. In particular, the authors study the effects of El Nino and la Nina events on economic growth, using well-established empirical methods from climate econometrics and a global panel of countries over the period 1961-2019. My feeling is that the paper has potential, but the authors should convincingly address some key issues, which I think currently undermine the validity and innovativeness of the study.

We thank the reviewer for the positive and encouraging comments.

First, a suggested interpretation of the estimated effects is grossly missing. The authors estimate economic losses from ENSO by means of an econometric model relating country-specific economic growth rates to an ENSO index, country-specific temperatures, and country-specific precipitation (with multiple powers and lags). They then use the marginal estimated effects to compute losses stemming from ENSO oscillations (please, pay attention to my third point below). However, they are silent on what mechanisms could drive economic losses from El Nino/La Nina. I feel this is relevant, especially when the effect of local weather change is controlled for. Intuitively, one would expect that El Nino events cause changes in the experienced weather conditions in different countries, which - in turn - affect local economic activity and production. However, given the model estimated by the authors, such effects should be largely captured by the precipitation and temperature variables. Hence, the “additional effect” of ENSO is very interesting and surprisingly large, but – for this to be credible - I feel the authors should make an interpretative effort, suggesting what mechanisms may lie behind the statistical relationship the paper seems to spot robustly. This is even more relevant as the present study reports much larger losses than in previous literature.

We thank the reviewer for the suggestion. Some previous studies reported immediate observable losses from ENSO events (for example, Changnon 1999; Zong et al. 2000), but there are losses not directly observable or as a delayed response to the initial climate shock. For example, in El Niño-affected regions (such as Australia), extended drought and forest fires cause not only infrastructure damage, loss of cattle stocks, and reduction in farming outputs, but also unemployment of people working in the sector (Vos 1999). These impacts lead to high inflation and reduced purchasing power with an economic effect transcending the affected workers and communities (Brunner 2002; Cashin et al. 2017). Through the high inflation, low purchasing power, and the associated low consumer confidence, the impact spills over to other sectors such as service and tourism of the broader economy and delays major investment and recruitment decisions. In addition, through global trades, the reduced agriculture and cattle production leads to higher commodity prices, affecting global economy. Thus the impact is not limited to the immediate tangible losses, but far larger, intangible and long-lasting.

We have now included more discussion in Line 144-150.

References:

- Brunner, A. D. (2002). El Nino and world primary commodity prices: warm water or hot air?. *Review of Economics and statistics*, 84(1), 176-183.
- Cashin, P., Mohaddes, K., & Raissi, M. (2017). Fair weather or foul? The macroeconomic effects of El Niño. *Journal of International Economics*, 106, 37-54.

Changnon, S. A. (1999). Impacts of 1997–98 El Niño–generated weather in the United States. *Bulletin of the American Meteorological Society*, 80(9), 1819-1828.

Vos, R., Velasco, M., & Edgar de Labastida, R. (1999). Economic and social effects of El Niño in Ecuador, 1997-1998. Washington, DC: Inter-American Development Bank.

Zong, Y. & Chen, X. The 1998 flood on the Yangtze, China. *Natural Hazards* 22, 165-184 (2000).

Second, the innovativeness of the paper should be streamlined much more effectively. The authors sometimes seem to motivate their study on the basis of inconclusive and scant evidence about the effects of El Niño/La Niña on the macroeconomy (see e.g. the abstract at lines 18-19). While I agree that further studies are definitely necessary, I have some issues grasping the novelty of the present paper with respect to the literature. For example, Smith and Ubilava (2017, GEC) - which the authors cite - offer a quite similar approach, finding non-linear effects, spanning across different years and varying across countries and climatic zones. Cashin et al. 2017 (2017, JIntEco) - which the authors cite - offer a much more comprehensive assessment of the macroeconomic effects of El Niño and La Niña events, though with a different methodology. I feel the authors should make crystal clear the piece of evidence they add to the literature.

Our nonlinear model is different in some key characteristics from previous approaches such as using a Heaviside model to depict the nonlinear impact. Our fully nonlinear model, which shares the feature of an asymmetric impact between El Niño and La Niña, allows the impact to increase exponentially with El Niño amplitude, which is assumed to be linearly proportional to El Niño amplitude in a Heaviside model. Additionally, our model allows the La Niña impact to change sign from a benefit during moderate La Niña to a damage during extreme La Niña, instead of assuming one sign of impact for all La Niña events in a Heaviside model. Our fully nonlinear model is more realistic. Allowing the impact to change nonlinearly with El Niño or La Niña amplitude is important for assessing ENSO impact on macroeconomy in a warming climate, given that ENSO amplitude is projected to increase.

My reading is that the originality lies in i. the role of teleconnections (which would deserve more space in the paper), ii. the additional effect of ENSO beyond that of local temperature and precipitations (see my point above), iii. the projected effects on growth, which are very large and need to be robust (see my third and fourth points immediately below).

We thank the reviewer for recognizing our points and for the suggestions to improve. We have added the following discussion.

- i. In terms of the role of country-specific teleconnection, we stress that our inclusion of interaction between common ENSO shock with country-specific teleconnection in the econometric model is to test the heterogeneity of ENSO impact across individual countries. We find that the inclusion only makes a minor and insignificant change to the common ENSO impact; see our detailed response to your third comment below.
- ii. In terms of the effect of ENSO beyond that of local weather anomalies, please see our detailed response to your first comment, we have stressed this point.
- iii. In terms of the large effect on growth, we thank you for recognizing this aspect. Please see our response to your third and fourth comments regarding to the robustness of our projection.

Third, the authors estimate a growth effect while they mostly report results in terms of a level effect. While the paper finds a robust non-linear effect of ENSO on growth, economic losses are mostly reported in terms of trillions of dollars of economic damage, which is suggestive of a level effect. I feel this approach is problematic for two reasons: first, it does not account for the debate on the persistence of climate impacts on the macroeconomy (see e.g. Diaz and Moore 2017, NCC; Bastien-Olvera et al. 2021, ERL);

We should have made it clear that this is not the case. We introduced multiple lags in our econometric model to test whether and how ENSO affects the level or/and the growth effect. If ENSO only affects the level of GDP, the contemporaneous and lagged effects of ENSO would be opposite in sign. That is, a negative impact in the occurrence year of ENSO would be followed by a positive “rebound” in the following year, leading to a sum of contemporaneous and lagged effects (i.e., cumulative effect) close to zero.

Instead, we find substantial and significant lagged effects of ENSO, which have the same sign as the contemporaneous effect, and the magnitude is greater than the level effect (**Table S1**). This suggests an accelerated growth effect of ENSO after the initial shock. In our estimates, such as that presented in **Fig. 2a**, the losses mostly come from growth effect. Our result also supports the persistence of ENSO impact on the macroeconomy.

second, it makes results more difficult to interpret from a quantitative standpoint. Indeed, 1 trillion USD is relatively low for the US economy, while extremely large for a small African country; similarly, it is difficult to say whether a loss of 2.1 trillion (a figure found in the abstract) is large or small for the global economy without a meaningful term of comparison (e.g. the value of the global GDP in the year of the shock or the loss generated by another event, e.g. the global financial crisis).

We have also estimated economic losses separately in different groups of countries, to show the heterogeneous response of different countries to ENSO (**Fig. 1**). For example, given the magnitude of 1997-98 El Niño (+2.27 s.d.), poor countries (such as small African countries) exhibit a loss of 6.08% while rich countries (including US) experience a loss of 3.28% (**Fig. 1d**). As our goal is to estimate ENSO’s global impact and its future change, we use all-country samples to train the econometric model for our main results in this paper.

We estimate that the cumulative loss of US\$1.3, US\$2.1 and US\$4.0 trillion from the 1982-83, 1997-98 and 2015-16 extreme El Niño events accounted for about 4% of global GDP in total over the four years (Line 156). Data from World Bank shows that the 2008 global financial crisis, for example, caused a cumulative loss in global GDP growth by ~5.7% (-2.3% from 2007 to 2008 and -3.4% from 2008 to 2009).

Further, while the authors account for inflation, it is not always clear in dollars of which year results are computed.

We have included that all the GDP data (including SSP projection) and estimated costs in this paper are in constant 2015 US\$.

More relevantly, I failed to get the exact meaning of equations 2 and 4. To my reading, they do not reflect the impact on growth the authors would like to measure. I would kindly encourage the authors to provide more details, possibly together with examples for some countries. Indeed,

it is not evident what losses in which countries are driving the global loss estimates the authors come up with. More clarity along these lines would make the results much more trustable. Unpacking the results across countries would offer an intriguing perspective, but this is just a suggestion.

We thank the reviewer for this suggestion. We have now provided more detailed steps of how the observed ENSO-induced economic loss, in-dollar value, is calculated (Line 486-490).

We have attempted to assess the country-specific estimate of economic loss by including an interaction of the common ENSO shock (Niño3.4) with country-specific teleconnection in the econometric model, similar to a previous study linking the common ENSO shock with local weather conditions (Generoso et al. 2020), as below:

$$\Delta \log(y_{it}) = \sum_{l=0}^n \{ \alpha_{1,l} E_{t-l} + \alpha_{2,l} E_{t-l}^2 + \gamma_{1,l} \psi_i E_{t-l} + \gamma_{2,l} (\psi_i E_{t-l})^2 + \beta_{1,l} T_{it-l} + \beta_{2,l} T_{it-l}^2 + \lambda_{1,l} P_{it-l} + \lambda_{2,l} P_{it-l}^2 \} + \mu_i + \theta_{1i} t + \theta_{2i} t^2 + \varepsilon_{it}$$

where $\psi_i E_{t-l}$ is the interaction term between the common ENSO shock and country-specific teleconnection. The estimated regression coefficients are listed below:

Table R2A. Same as Table S1, but with regression coefficients of interaction term (γ) added.

	Lag 0	Lag 1	Lag 2	Lag3	Lag4	Lag5
$\alpha_{1,0}$	-0.0032***	-0.0042***	-0.0052***	-0.0055***	-0.0049***	-0.0051***
$\alpha_{1,1}$		6.19×10^{-5}	-0.0011	-0.0015	-0.0016	-0.0020*
$\alpha_{1,2}$			9.26×10^{-4}	2.39×10^{-4}	1.06×10^{-4}	3.79×10^{-4}
$\alpha_{1,3}$				4.83×10^{-4}	-1.09×10^{-4}	3.63×10^{-4}
$\alpha_{1,4}$					0.0021**	0.0026***
$\alpha_{1,5}$						-0.0011
$\alpha_{2,0}$	3.93×10^{-4}	2.76×10^{-4}	5.03×10^{-4}	4.69×10^{-4}	7.77×10^{-5}	3.53×10^{-4}
$\alpha_{2,1}$		-0.0019***	-0.0023***	-0.0023***	-0.0022***	-0.0020***
$\alpha_{2,2}$			-0.0024***	-0.0027***	-0.0024***	-0.0025***
$\alpha_{2,3}$				-0.0012**	-0.0014**	-0.0015**
$\alpha_{2,4}$					-0.0012*	-8.26×10^{-4}
$\alpha_{2,5}$						9.36×10^{-4}
$\gamma_{1,0}$	2.05×10^{-4}	3.33×10^{-4}	2.36×10^{-4}	2.02×10^{-4}	8.63×10^{-5}	1.37×10^{-4}
$\gamma_{1,1}$		-4.35×10^{-4}	-4.76×10^{-4}	-6.55×10^{-4} *	-6.70×10^{-4} *	-5.82×10^{-4}
$\gamma_{1,2}$			-3.77×10^{-4}	-5.20×10^{-4}	-6.42×10^{-4} *	-6.94×10^{-4} *
$\gamma_{1,3}$				-8.35×10^{-4} **	-8.44×10^{-4} **	-9.62×10^{-4} **

$\gamma_{1,4}$					$-6.71 \times 10^{-4**}$	$-7.95 \times 10^{-4**}$
$\gamma_{1,5}$						5.86×10^{-5}
$\gamma_{2,0}$	-2.80×10^{-5}	-2.07×10^{-5}	-2.18×10^{-5}	-2.48×10^{-5}	-2.12×10^{-5}	-3.03×10^{-5}
$\gamma_{2,1}$		3.39×10^{-5}	3.56×10^{-5}	2.87×10^{-5}	2.34×10^{-5}	1.57×10^{-5}
$\gamma_{2,2}$			-3.67×10^{-7}	2.91×10^{-6}	-6.85×10^{-6}	-3.83×10^{-6}
$\gamma_{2,3}$				5.04×10^{-6}	6.74×10^{-6}	1.17×10^{-5}
$\gamma_{2,4}$					1.45×10^{-5}	6.85×10^{-6}
$\gamma_{2,5}$						-3.43×10^{-5}
R^2	0.1973	0.2006	0.2033	0.2055	0.2068	0.2075

As can be seen from **Table R2A**, coefficients of the interaction terms show little statistical significance. We compare the estimated global losses of observed extreme El Niño events. The result from the model with interaction terms doesn't show a significant difference from our original result (**Fig. R2A**). However, the model provides information on country-specific impacts. We have calculated country-level in-dollar-value losses from 2015-16 extreme El Niño. As can be seen from **Fig. R2B**, countries with the greatest economy (like US and China) contribute to much of loss, despite a smaller GDP growth loss, as expected.

Fig. R2A. Cumulative effect of three major extreme El Niño events in 1982/83 (yellow), 1997/98 (red) and 2015/16 (purple) on the global total GDP under (a) the original econometric model (same as Fig. 2a), and (b) new econometric model with interaction terms added. Shadings indicate the 95% confidence level for each event based on a Bootstrap method.

Fig. R2B. Country-level cumulative losses from 2015-16 extreme El Niño under the country-specific econometric model. The map is created in the MATLAB computing environment using the M_Map mapping package (Pawlowicz, R., 2020. "M_Map: A mapping package for MATLAB", version 1.4m, [Computer software], available online at www.eoas.ubc.ca/~rich/map.html).

Fourth. In their projection exercises, it is unclear what drivers of growth the authors consider and what they do not. For example, in the seminal paper by Burke et al. (2015, Nature) projections include socio-economic trends (proxied by historical growth rates of GDP). I have the impression the authors exclude such trends when projecting the GDP of various countries. For example, this seems to be the case in the section “Substantial additional loss in economic production from increased ENSO amplitude” (from line 197). Accounting for such trends may change the projection results sensibly. Similarly, accounting for future temperature will affect growth rates; however, the counterfactual scenario seems to assume that ENSO will change alongside future climatic and SSP scenarios while temperatures are not. I think the authors should either change their projections or offer a crystal-clear and convincing explanation of their counterfactual and projection exercises.

We followed Burke et al. (2015 & 2018) to train our econometric model by fitting with historical climate and economic data. The socio-economic linear and quadratic trends, were included as $\theta_{1i}t$ and $\theta_{2i}t^2$ in our model (Line 94-96 and 469-471).

Further, we assume that the future increase of temperature doesn't change the nonlinear ENSO impact. Future loss from mean temperature warming could be estimated using the historical response function as Burke et al. (2018), but that is not the goal of our work.

The estimated loss using the econometric model is dependent on the coefficients $(\alpha_{1,i}, \alpha_{2,i})$, (which are independent from rising temperature) and the projected future GDP. Two ENSO time series are constructed, one based on the projected ENSO time series, and the other on scaled projected time series such that ENSO variability is the same as in the model historical period. Using each of these ENSO time series as input, we obtain two time series of future ENSO impact on GDP growth rate, which are used to modify the projected growth rate. From the two time series of modified growth rate and the projected, we calculate two new time series of GDP. The difference is attributed as due to changes in ENSO.

Fifth, aggregation with discount rates may be problematic. Impacts on economic production are aggregated with discount rates chosen by the authors. Figure 4 is the most communicative example. I find this choice debatable. First, the last two decades have witnessed a fierce debate

about the selection of discount rates, and results from impact studies typically showed large sensitivity to such a parameter. This paper is not an exception.

We agree. We used a mid-range discount rate as an example to gauge the economic damages from changing ENSO in dollar value that is comparable across years during the long period. Choice of discount rate is itself a source of uncertainty, and this is why we use a range of discount rate (1-5%) values.

The uncertainty from discount rates, while substantial, is relatively small compared with uncertainty from that projected ENSO change – which we show to be the dominant source (Fig. 4f and Line 563-586). We also provide the result in terms of percentage GDP loss without involving a discount rate (Fig. 3), see our response below.

Second, it is not clear to me why the authors need to discount back in time a projected impact. For example, they could just compute the loss as the ratio between the projected global GDP accounting for ENSO effects and the projected GDP without the ENSO effect. This would be more similar, for example, to the approach by Burke et al. 2015 and to the majority of damage functions.

You are right. We have calculated the projected loss equivalent to a ratio of projected global GDP in 2099. The multi-model ensemble mean ratio between the projected GDP accounting for ENSO effects and the projected GDP without the ENSO effect is 1.14, indicating a cumulative 14% reduction of global GDP by 2099. Moreover, we find that the ratio of projected GDP increases nonlinearly with the ratio of projected ENSO amplitude (Fig. R2C), similar to the relationship in Fig. 4e. These conclusions are independent of an assumed discount rate or a projected GDP value.

Fig. R2C. Nonlinear relationship between ratios of Niño3.4 variability (21st / 20th century) and ratios of projected global GDP in 2099 (without projected ENSO's effect / without counterfactual ENSO's effect) from CMIP6 models under four IPCC scenarios. Different colors refer to four IPCC scenarios. The R square and P value of a nonlinear fit are given.

Finally, let me provide few other yet less substantive comments.

• at lines 95-98 the authors state they include mean temperature and precipitation in their model to estimate the effects of mean state change and introduce the assumption that ENSO-induced temperature and precipitation changes are independent of mean state changes. First, I am not convinced that equation 1 reflects this assumption (rather, it reflects yearly deviation from country-specific trends that may be caused by any events, including ENSO); second, I am not convinced the assumption of independence is realistic and/or innocent.

We have tested whether ENSO variation is independent of temperature and precipitation by removing the ENSO signal from country-level mean temperature and precipitation, and then re-estimate the econometric model. As can be seen in **Table R2B**, there is little difference in the coefficients of ENSO impact ($\alpha_{1,l}$, $\alpha_{2,l}$), mean temperature and precipitation effect ($\beta_{1,l}$, $\beta_{2,l}$ and $\lambda_{1,l}$, $\lambda_{2,l}$), suggesting our results remain virtually the same after excluding ENSO signal from the mean state.

Table R2B. Comparison of regression coefficients of Niño3.4 index, mean temperature and precipitation in the original econometric model (left) and the econometric model with ENSO removal in the mean temperature and precipitation change.

	Lag 3, original	Lag 3, ENSO removed in mean state change
$\alpha_{1,0}$	-0.0055***	-0.0051***
$\alpha_{1,1}$	-0.0027***	-0.0029***
$\alpha_{1,2}$	-6.20×10^{-4}	-3.41×10^{-4}
$\alpha_{1,3}$	-9.73×10^{-4}	-8.25×10^{-4}
$\alpha_{2,0}$	2.92×10^{-4}	3.08×10^{-4}
$\alpha_{2,1}$	-0.0021***	-0.0021***
$\alpha_{2,2}$	-0.0026***	-0.0026***
$\alpha_{2,3}$	-0.0012**	-0.0012**
$\beta_{1,0}$	0.0125***	0.0123***
$\beta_{1,1}$	-0.0076**	-0.0071**
$\beta_{1,2}$	-0.0013	-0.0015
$\beta_{1,3}$	-0.0023	-0.0016
$\beta_{2,0}$	-3.68×10^{-4} ***	-3.48×10^{-4} ***
$\beta_{2,1}$	2.31×10^{-4} **	2.31×10^{-4} **
$\beta_{2,2}$	7.33×10^{-5}	1.16×10^{-4}
$\beta_{2,3}$	-2.95×10^{-5}	-7.23×10^{-5}
$\lambda_{1,0}$	-3.48×10^{-4}	-4.84×10^{-4}

$\lambda_{1,1}$	-0.0010	-2.11×10^{-4}
$\lambda_{1,2}$	-3.51×10^{-4}	1.95×10^{-5}
$\lambda_{1,3}$	-0.0042	-0.0049
$\lambda_{2,0}$	-4.26×10^{-5}	-4.48×10^{-5}
$\lambda_{2,1}$	3.79×10^{-5}	3.84×10^{-6}
$\lambda_{2,2}$	-4.79×10^{-5}	-6.42×10^{-5}
$\lambda_{2,3}$	2.50×10^{-4}	2.88×10^{-4}
R^2	0.2041	0.2040

- the effects of temperature and precipitations are not discussed and it is also not clear whether these variables are included in the estimation in their population-weighted version or not. This may have implications for how the authors capture exposure of economic production to the weather and for the interpretation of results.

What we have shown is after mean temperature and precipitation are already included. Here the mean temperature and precipitation are also population-weighted into country-average. We have now added the discussion about effect of temperature and precipitation. Similar to findings of Burke et al. (2015), the mean temperature affects economy nonlinearly, while mean precipitation shows little and insignificant impact.

- The links with the blossoming climate econometrics literature could be better surveyed and the paper better positioned in such a stream of studies. For example, El Nino and La Nina are known to exert impacts through changes in precipitation patterns (among other channels). Palagi et al. (2022, PNAS) and Kotz et al. (2022, Nature) are two extremely recent contributions emphasizing the impact of precipitation anomalies on growth and the macroeconomy. The authors could leverage these results to provide a more informative contextualization of their findings. This links back to my first point.

Great point. ENSO affects macroeconomy directly through local weather conditions such as temperatures and precipitation. We have provided the context that ENSO is a recognized major driver of weather extremes in many affected regions, and cited the economic impacts revealed by previous investigation (for example, temperature extremes: Callahan and Mankin, 2022; precipitation extremes: Palagi et al. 2022; Kotz et al. 2022). See Line 47.

Thank you again for your helpful and encouraging comments, which have guided us to improve the paper a lot.

Response to Reviewer #3

This article, which aims to assess the overall economic cost of ENSO in a context of climate change, covers an important subject area. My comments focus on how the ENSO impact is considered and the relevance of an average assessment of its cost in terms of overall GDP (knowing that I am not a climatologist). I find many issues that must be addressed for the assessment to be credible.

My comments relate to the estimation of the country-specific teleconnection strengths as well as the estimation of the economic cost of ENSO-related events. It seems fundamental to me to solidify these results before any climate projection.

We thank the reviewer for the thorough and constructive comments, which improve our work. We provide our response below.

1. ENSO index and country-specific teleconnection

The authors propose a measure of the degree of teleconnection specific to each country. More detail should be given regarding the construction of the indicator reflecting the magnitude of teleconnection patterns.

We have now provided more detailed steps for the construction of ENSO country-specific teleconnection.

The estimation of the regression coefficients ρ and τ raises questions in the absence of a clearly specified model. Are the temperature and precipitation anomalies, used as explained variables, standardized?

The Niño3.4 index, temperature and precipitation anomalies are standardized in our study. The normalization ensures that the regression coefficients τ and ρ are dimensionless, and comparable. We have made clear in the revised manuscript (Line 442-443).

Are precipitation included as a covariate in the regression when analyzing temperature and vice versa to control for the correlation between temperature and precipitation.

We have now used partial regression instead to calculate the regression coefficient of temperature onto Niño3.4 index when removing the precipitation impact, and vice versa (Line 446-448).

Moreover, several factors can lead to a waxing and waning in the strength of the ENSO teleconnections: changes in the evolution of the SST pattern during ENSO, changes in the ENSO's temporal behavior, changes in the mean base climatology, and its modulation of the teleconnection process, and simple sampling variations. It could then be helpful to review the robustness of the country-specific teleconnection metric assessed by the authors by analyzing its temporal stationarity throughout the entire reference period.

We thank the reviewer for the suggestion. First, to test spatial variability of ENSO SST anomaly that may affect teleconnection, we apply a 36-year running window regression of DJF SST anomalies onto DJF Niño3.4 (1950-1985, 1951-1986, ... 1986-2021, a total of 37 windows), and then compare the pattern correlation of the 37 samples with the overall regressed

SST anomalies in the tropical Pacific (**Fig. R3A-a**). As can be seen in **Fig. R3A-b**, there is little significant change in ENSO SST pattern in different time periods, suggesting the ENSO SST pattern is rather stationary.

Second, to test temporal change of ENSO teleconnection, we apply a 36-year running window regression of May(0) to April(1) monthly surface air temperature and precipitation anomalies onto DJF Niño3.4 (1950-1985, 1951-1986, ... 1986-2021), to construct 37 samples of cumulative temperature and precipitation teleconnection strength (τ and ρ) in the globe. We compare the pattern correlation of these 37 samples with the overall cumulative teleconnection (**Fig. R3B-a, b**). As can be seen in **Fig. R3B-c, d**, the global teleconnection patterns of both temperature and precipitation are stationary.

Fig. R3A. a, Regressed DJF SST anomalies onto Niño3.4 index (normalized) in the tropical Pacific during 1950-2021. Only the regression coefficients above 95% significance level are shown as shading. The map is created in the MATLAB computing environment using the M_Map mapping package (Pawlowicz, R., 2020. "M_Map: A mapping package for MATLAB", version 1.4m, [Computer software], available online at www.eoas.ubc.ca/~rich/map.html) . **b**, Pattern correlation of regressed SST anomalies from 37 samples, with the overall regressed SST anomalies during 1951-2021.

Fig. R3B. **a**, Global cumulative ENSO teleconnection of temperature (τ), calculated as the sum of May(0) to April(1) monthly regression coefficients of surface air temperature anomalies onto DJF Niño3.4 (both normalized) during 1950-2021. **b**, Same as **a**, but for precipitation (ρ). **c**, Pattern correlation of cumulative ENSO teleconnection of temperature (τ) from 37 samples, with the overall one during 1951-2021. **d**, Same as **c**, but for precipitation (ρ). The maps were created in the MATLAB computing environment using the M_Map mapping package (Pawlowicz, R., 2020. "M_Map: A mapping package for MATLAB", version 1.4m, [Computer software], available online at www.coas.ubc.ca/~rich/map.html).

In line 403, the authors state that the estimated coefficients are weighted by population density in 2020. First, what is the source of the population density data?

We included this information. The population density data is from Gridded Population of the World (GPW) (Line 410-412).

Second, the authors should give better justification with regard to this choice concerning the spatial aggregation procedure. Number of economic studies aggregate ENSO teleconnections within countries borders using a fixed set of population weights, so that the relevant concept is the average weather experienced by a person in the administrative area, not the average weather experienced by a place. However, this procedure implies that the aggregation gives a higher weight to highly populated (urban) areas while areas characterized by low population densities may play a major economic role and while being impacted by ENSO shocks. This is particularly the case for agricultural regions, a sector through which the majority of ENSO-related economic shocks spread.

We thank the reviewer for raising this point. Because ENSO affects economy not only in the agriculture sector (though agriculture is one of the most affected components), but also many other economic activities such as tourism, trade and services, which are labor-related and a key part of the economy. The spill-over effect can be far greater than the direct and tangible loss. We average the climate shock by population density only to gauge the impact sensitivity to varying groups of economies. Ultimately when all countries are included, as the case in our study, the impact does not involve a country-specific depiction of ENSO teleconnection.

In line 405, the authors divide the countries into two categories based on the accumulation of the country-specific temperature and precipitation teleconnection strengths (ψ_i) so that countries where $\psi_i < 0.5$ are weakly connected and countries where $\psi_i > 0.5$ are connected. This categorization is of particular concern to me. On what basis is this classification made? It seems from Figure S4 that the choice of such a low threshold value completely crushes the country heterogeneity of ENSO shocks and that almost all countries belong to the category of teleconnected countries. In other words, what is the point of calculating a specific indicator for each country to ultimately arrive at such a broad categorization?

A specific threshold for distinguishing “teleconnected” and “weakly-teleconnected” countries is to separately estimate the nonlinear economic impact from these two groups of countries. We have applied a sensitivity test to gauge whether the choice of ψ threshold may affect our result. For example, by using $\psi > 1$ or $\psi > 1.5$ instead, there is no significant difference in the response function (**Fig. R3C**).

Fig. R3C. Global nonlinear relationship between D(0)JF(1) Niño3.4 index (normalized) and 3-year cumulative (from year 0 to year 3) change in log GDP per capita for teleconnected countries with threshold of $\psi > 1.5$ (brown curve), $\psi > 1$ (red curve), and $\psi > 0.5$ (yellow curve). Shading indicating the 95% confidence level based on a Bootstrap method.

We agree with the reviewer that a low threshold ($\psi > 0.5$) allows most of countries belong to the category of “teleconnected”. Hence, we choose $\psi > 1$ as the new threshold to make a more balanced distinction for the two groups (**Fig. S5b**).

Fig. S5. **b**, Classification of countries into groups of “teleconnected” defined as $\psi > 1$ (red), and “weakly-teleconnected” defined as $\psi \leq 1$ (blue). The map is created in the MATLAB computing environment using the M_Map mapping package (Pawlowicz, R., 2020. "M_Map: A mapping package for MATLAB", version 1.4m, [Computer software], available online at www.eoas.ubc.ca/~rich/map.html).

It seems to me, moreover, that this methodological choice is decisive for the rest of the article and therefore the determination of the overall cost of ENSO.

We should have made it clear that this is not the case. We examined the heterogeneity effect only in this section only as a gauge of the impact sensitivity to grouping of countries. Subsequently, we have included all countries, which does not involve a country-specific depiction of ENSO teleconnection. As our ultimate goal is to estimate the global economic impact of ENSO shocks and its future change, all of our main results hereafter are generated from the climate-economy model that is fitted by data from all countries, involving no country-specific depiction of ENSO teleconnection.

It would be necessary to consider alternative global measurements of the level of teleconnection based on more precise measurements (spatial variability, spatial extent, etc.) and to accompany the results with a map showing the level of teleconnection by grid point (before the aggregation at the country level).

In the context of sensitivity tests (see above), we have now given a grid-point temperature and precipitation teleconnection map separately in **Fig. S4**.

Fig. S4. Global cumulative grid-point ENSO teleconnection of **a** temperature (τ) and **b** precipitation (ρ), calculated as the sum of May(0) to April(1) monthly regression coefficients of normalized surface air temperature and precipitation anomalies onto normalized DJF Niño3.4. The maps were created in the MATLAB computing environment using the M_Map mapping package (Pawlowicz, R., 2020. "M_Map: A mapping package for MATLAB", version 1.4m, [Computer software], available online at www.eoas.ubc.ca/~rich/map.html).

2. Empirical econometric model

The authors test a distributed-lag timeseries model to estimate the economic cost of ENSO events. Is this model estimated by the GLS or by the OLS?

Our model is estimated by the Ordinary Least Squares (OLS). We have now included it (Line 460).

I think it is essential to have more information on the possible presence of multicollinearity (using the VIF for example). Indeed, the measurement of ENSO shocks being indexed only by time (shock common to all countries), the risk of quasi collinearity with the country fixed effects (μ_i) is high and likely to bias the estimates.

There is little collinearity of ENSO shocks with a country-fixed effect represented by country-specific ENSO teleconnection. Taking the teleconnection as a part of characteristics for individual countries, we examine the correlation between country-specific teleconnection (ψ_i) and country-fixed effect (μ_i) in our econometric model. As can be seen in **Fig. R3D**, there's no evidence of a significant collinearity between them.

Fig. R3D. Relationship between the country-specific ENSO teleconnection (ψ_i , x-axis) and country-fixed effect (μ_i , y-axis) extracted from the econometric model. The very low correlation (0.02) indicates little collinearity between them.

Using a common ENSO shock to estimate its overall cost does not seem to constitute a significant contribution compared to the previous literature on the subject which explicitly introduces a measure of the strength of the teleconnection specific to each country in the econometric specification or via the interaction between the common shock and country specific weather anomalies. Why not integrate the results from the calculation of country-specific teleconnections within the econometric specification? Indeed, the distinction made in Figure 1 does not seem to show any difference between the categories of countries and seems insufficient.

We thank the reviewer for the suggestion. To test this hypothesis, we have introduced the interaction terms to assess how ENSO affects economy under different teleconnection strengths.

This approach is similar to a previous study that linking the common ENSO shock with local weather conditions (Generoso et al. 2020). The econometric model is therefore written by

$$\Delta \log(y_{it}) = \sum_{l=0}^n \{ \alpha_{1,l} E_{t-l} + \alpha_{2,l} E_{t-l}^2 + \gamma_{1,l} \psi_i E_{t-l} + \gamma_{2,l} (\psi_i E_{t-l})^2 + \beta_{1,l} T_{it-l} + \beta_{2,l} T_{it-l}^2 + \lambda_{1,l} P_{it-l} + \lambda_{2,l} P_{it-l}^2 \} + \mu_i + \theta_{1i} t + \theta_{2i} t^2 + \varepsilon_{it}$$

where $\psi_i E_{t-l}$ is the interaction term between the common ENSO shock and country-specific teleconnection. The estimated regression coefficients are listed below:

Table R3A. Same as Table S1, but with regression coefficients of interaction term (γ) added.

	Lag 0	Lag 1	Lag 2	Lag3	Lag4	Lag5
$\alpha_{1,0}$	-0.0032***	-0.0042***	-0.0052***	-0.0055***	-0.0049***	-0.0051***
$\alpha_{1,1}$		6.19×10^{-5}	-0.0011	-0.0015	-0.0016	-0.0020*
$\alpha_{1,2}$			9.26×10^{-4}	2.39×10^{-4}	1.06×10^{-4}	3.79×10^{-4}
$\alpha_{1,3}$				4.83×10^{-4}	-1.09×10^{-4}	3.63×10^{-4}
$\alpha_{1,4}$					0.0021**	0.0026***
$\alpha_{1,5}$						-0.0011
$\alpha_{2,0}$	3.93×10^{-4}	2.76×10^{-4}	5.03×10^{-4}	4.69×10^{-4}	7.77×10^{-5}	3.53×10^{-4}
$\alpha_{2,1}$		-0.0019***	-0.0023***	-0.0023***	-0.0022***	-0.0020***
$\alpha_{2,2}$			-0.0024***	-0.0027***	-0.0024***	-0.0025***
$\alpha_{2,3}$				-0.0012**	-0.0014**	-0.0015**
$\alpha_{2,4}$					-0.0012*	-8.26×10^{-4}
$\alpha_{2,5}$						9.36×10^{-4}
$\gamma_{1,0}$	2.05×10^{-4}	3.33×10^{-4}	2.36×10^{-4}	2.02×10^{-4}	8.63×10^{-5}	1.37×10^{-4}
$\gamma_{1,1}$		-4.35×10^{-4}	-4.76×10^{-4}	-6.55×10^{-4} *	-6.70×10^{-4} *	-5.82×10^{-4}
$\gamma_{1,2}$			-3.77×10^{-4}	-5.20×10^{-4}	-6.42×10^{-4} *	-6.94×10^{-4} *
$\gamma_{1,3}$				-8.35×10^{-4} **	-8.44×10^{-4} **	-9.62×10^{-4} **
$\gamma_{1,4}$					-6.71×10^{-4} **	-7.95×10^{-4} **
$\gamma_{1,5}$						5.86×10^{-5}
$\gamma_{2,0}$	-2.80×10^{-5}	-2.07×10^{-5}	-2.18×10^{-5}	-2.48×10^{-5}	-2.12×10^{-5}	-3.03×10^{-5}
$\gamma_{2,1}$		3.39×10^{-5}	3.56×10^{-5}	2.87×10^{-5}	2.34×10^{-5}	1.57×10^{-5}
$\gamma_{2,2}$			-3.67×10^{-7}	2.91×10^{-6}	-6.85×10^{-6}	-3.83×10^{-6}
$\gamma_{2,3}$				5.04×10^{-6}	6.74×10^{-6}	1.17×10^{-5}

$\gamma_{2,4}$					1.45×10^{-5}	6.85×10^{-6}
$\gamma_{2,5}$						-3.43×10^{-5}
R^2	0.1973	0.2006	0.2033	0.2055	0.2068	0.2075

As can be seen from **Table R3A**, coefficients of interaction term show little statistical significance. We further compare the estimated global losses of observed extreme El Niño events. The result from the model with interaction terms doesn't show significant difference with original result (**Fig. R3E**), suggesting the interaction between common ENSO shock with country-specific teleconnection has a minor effect.

Fig. R3E. Cumulative effect of three major extreme El Niño events in 1982/83 (yellow), 1997/98 (red) and 2015/16 (purple) on the global total GDP under **a** original econometric model (same as Fig. 2a), and **b** new econometric model with interaction term added. Shadings indicate the 95% confidence level for each event based on a Bootstrap method.

The distinction between rich and poor countries should also be clarified. On what basis is this distinction made?

Here we classify the rich and poor countries following the distinction of Burke et al. (2015), that is, rich (poor) countries are those with purchasing-power-parity-adjusted (PPP) per capita income in 1980 above (below) the median. We have provided the details in the revised version.

This distinction may seem tautological from an economic point of view, as poor countries are more vulnerable to climatic hazards...

Great point. This is what we have stressed. Poor countries show larger nonlinear response to ENSO mainly because they happen to be more strongly affected by ENSO (histograms of Fig. 1d) (Line 135-136).

A more relevant distinction should be made on the basis of the countries structural characteristics by distinguishing, for example, countries dependent on agriculture, countries whose economic fabric is more diversified and thus evaluating the heterogeneity estimated cost of ENSO events.

We thank the reviewer for the suggestion. We have additionally estimated the response of agriculture-dependent countries (defined by GDP share of agriculture greater than 20%), whose

economy is more likely affected by weather extremes from ENSO teleconnection. As expected, agriculture-dependent countries exhibit larger response to El Niño than agriculture-independent countries. We have included this result in the new **Fig. 1c**.

Fig. 1. c, Global nonlinear relationship between D(0)JF(1) Niño3.4 index (normalized) and 3-year cumulative (from year 0 to year 3) change in log GDP per capita for agriculture-dependent (green curve) and agriculture-independent countries (blue curve) during 1960-2019, with shading indicating the 95% confidence level based on a Bootstrap method. Histograms below show the distribution of country-specific ENSO teleconnection strength for agriculture-dependent (green bars) and agriculture-independent (blue bars) countries, respectively.

The annual measurement of the ENSO indicator (Nino 3.4) should be subject to robustness. The authors should consider comparing their non-linear (quadratic) measurement to a measurement based on event occurrence (based on threshold values) thus justifying the methodological contribution of the paper with respect to the prior literature relying on a Heaviside function in order to dissociate El Niño events from La Nina events. It is important to note that the references mentioned by the authors use both a Heaviside function and a quadratic specification in order to characterize ENSO events within the framework of their econometric specifications.

To compare our result with previous estimates, we have built a similar Heaviside model with different lags according to prior studies (Smith and Ubilava, 2017; Generoso et al. 2020):

$$\Delta \log(y_{it}) = \sum_{l=0}^n \left\{ I(.)\alpha_{1,l}E_{t-l} + [1-I(.)]\alpha_{2,l}E_{t-l} + \beta_{1,l}T_{t-l} + \beta_{2,l}T_{t-l}^2 + \lambda_{1,l}P_{t-l} + \lambda_{2,l}P_{t-l}^2 \right\} + \mu_i + \theta_{1i}t + \theta_{2i}t^2 + \varepsilon_{it}$$

$$I(.) = \begin{cases} 1 & E \geq 0 \\ 0 & E < 0 \end{cases}$$

where $I(.)$ is the Heaviside indicator to determine the ENSO regimes. We therefore train the Heaviside model and estimate the global losses from observed extreme El Niño events. **Fig R3F** shows consistent results between these two models, both of which indicate a negative impact from El Niño but insignificant for La Niña, and estimate similar losses from extreme El Niño events in the historical record.

Fig. R3F. **a**, Same as Fig. 1a. **c**, Same as Fig. 2a. **b** and **d**, Same as **a** and **c**, but for the results from the Heaviside model.

Although our estimated loss from ENSO during the historical period is, mostly, consistent with prior studies in the historical estimate, we use an index-based fully nonlinear model, instead of a threshold-based Heaviside model, to assess future projection under climate change mainly for two reasons. First, threshold-based model is sensitive to the threshold value, which introduces potential uncertainty in the projected loss from ENSO events. Second, El Niño's impact increases nonlinearly with amplitude, which however is assumed to be linearly proportional to amplitude in a Heaviside approach; for La Niña, the impact changes sign from a benefit for small-amplitude events to a damage for large-amplitude events, but is assumed to be one sign for all events in a Heaviside model. One additional advantage is that index-based model is able to construct a continuous counterfactual ENSO and GDP timeseries, which allows us estimate continuously the projected loss under climate change.

Although the measure of the climate can reasonably be considered exogenous (not invalidating the hypotheses of the model), it would seem relevant to include control variables reflecting the characteristics of the countries such as the share of GDP in the Value added Agricultural, the degree of trade openness.

Effects that controlling characteristics of individual countries, such as share of GDP in agriculture, trade openness, are involved in the country-fixed effect μ_i , and in projected GDP for each country.

We thank you again for your helpful comments, which have guided us to explore deeper, and make our key points clearer. The paper is so much the better.

REVIEWER COMMENTS

Reviewer #1 (Remarks to the Author):

The authors have responded to my original comments and suggestions. Those responses are satisfactory. I still find it troubling that the effect of ENSO events is largely homogenous around the globe. But I can live with that. My remaining comments are as follows:

- are the results sensitive to omitting some years from the data? For instance, the authors could do a variant of rolling- or expanding-window regressions. Say, run the regression using 1960-2000 data; then 1961-2001 data, and so on. More broadly, the authors may consider some variant of cross-validation -- drop a bunch of random years, re-estimate the coefficients, repeat the process many times, and present the distribution of the coefficients.
- are the results robust to falsification checks? For instance, use the future ENSO data. Say, regress growth in period t on ENSO in periods $(t+5, t+4, \text{ and } t+3)$. There should be no statistically significant coefficients associated with ENSO variables.

Reviewer #2 (Remarks to the Author):

I think the paper has improved and that some of my comments have been addressed rather convincingly by the authors. However, in my reading, a few outstanding issues remain, perhaps regarding my most serious concerns.

First. Despite some efforts at amending a few lines of the text, I feel the interpretation of results is still poor. Specifically, it is unclear what mechanism may drive the ENSO-related losses. My reading of the main econometric model of this study (Equation 1) is that it assumes and tests how GDP growth is affected by (i) an ENSO index and (iii) local climate/weather, beyond a series of controlling factors. Results show that ENSO exerts a non-linear effect on growth conditional on local weather conditions. I wonder how. The effect cannot go through changes in yearly temperature and precipitations – as sometimes seems to emerge from the text (e.g. “As the economic impact of ENSO is underpinned by the direct climate response to ENSO for individual countries”) since they are accounted for in the econometric model. This means that El Nino and La Nina induce impacts on the economy through other channels that are relevant to growth. For example, it might well be that one of these channels is an increase in wildfires (as in the example the authors use in their response to my comment) yet only conditional on the fact that yearly precipitation and temperatures do not already reflect such an increase. However, fires are just one among many possible candidates. Interpreting the effect is even more relevant as the authors find a little role for teleconnections.

Further, in response to the authors’ reply to my comment, I feel the present analysis cannot say whether the macroeconomic effects are due to sectoral spillovers/cascading effects or because the direct effect of ENSO is sufficiently large in certain regions to reflect on the macroeconomic statistics. However, I appreciate such further reasoning, which I think should be further expanded to convince the reader about the large magnitude of the effects.

Finally, on this point, I feel the authors should justify why an inverted U shape for the ENSO effect is reasonable. Specifically, why should an increase in the DJF Nino3.4 index be beneficial to growth up to a certain level? What is such a level? Is it reasonable? Is it robust across specifications? The figures provide little information as the ENSO index is normalized.

Second, I still have issues with the persistence of the effect the authors find. I fully agree that the impact they find in the analysis is persistent, adverse to growth in most cases, and going beyond a temporary shift in the level of output. This was motivating me to ask why they report losses in levels, which I think is misleading. Indeed, the results seem to indicate that, on average, a change in the ENSO index will lower the average growth rate of GDP, with the economy taking a few years to adjust

to the new long-term growth pattern. I fear the authors tend to interpret their econometric model as if it would estimate an impulse-response effect, but this is not the case if my reading is correct. In short, I am sympathetic to the growth-effect view expressed by the authors, while I think there is an excessive and incoherent effort at expressing "damages" in lost trillions in a certain number of years (which is reminiscent of a level effect).

Finally, let me flag three minor points.

- I feel a figure showing the geographical distribution and intensity of losses is worth space in the main text. I suggest discussing more/including Fig. R2B or similar in the main text.
- As far as the projection exercise is concerned, what growth rate of output (g_t) is used for future time periods (eq. 6).? Further, I still feel the difference between the projected GDP in a certain SSP and the historical growth rate of a country is NOT accounted for in the projections.
- Please clarify exactly how the ENSO signal has been removed from country-specific temperature and precipitations for the robustness exercise. Table R2B is relevant; it is not clear how this information is used in the text.

Reviewer #3 (Remarks to the Author):

I would like to extend my appreciation to the authors for the considerable enhancements made to the paper and for addressing earlier remarks. However, there remain a few aspects that necessitate further elucidation.

1 - The initial clarification pertains to the insensitivity of the results with respect to the psi threshold parameter. It would be valuable to understand the authors' interpretation of this seemingly paradoxical finding.

2 - The explanation regarding the weighting applied to meteorological variables (based on population density) could be more explicit. The estimated results remain stable across various classifications; however, the only notable distinction is tied to the proportion of agriculture in GDP (low/high). The authors should delve deeper into this outcome and provide a clearer interpretation. Identifying an economic cost without pinpointing the transmission channel seems problematic, particularly in a projection exercise context.

3 - It is crucial to confirm that the obtained results do not rely on the population-based weighting method. For instance, I would be interested to see if the estimated costs associated with the 2015/2016 El Niño event affect China (despite its weak teleconnection) and the United States similarly when using an alternative weighting approach.

4 - The authors have satisfactorily addressed the matter of ENSO temporal stability. It is worth noting, however, that the literature indicates that the occurrence of ENSO CP, EP, and mixed phenomena can considerably influence global temperatures and precipitation. My primary concern was that literature projections suggest a potentially higher frequency of CP and mixed-type events, which would have varying impacts on temperatures and precipitation.

5 - The provided response to the inquiry about introducing control variables appears inadequate. Indeed, individual fixed effects only account for the impact of time-invariant factors, and incorporating variables such as trade, the proportion of agriculture, development, and financial depth is essential, as it could modify the estimated coefficients associated with ENSO effects. Failing to do so poses a significant risk of omitted variable bias.

We thank the reviewers for their positive and helpful comments.

Response to Reviewer #1

The authors have responded to my original comments and suggestions. Those responses are satisfactory. I still find it troubling that the effect of ENSO events is largely homogenous around the globe. But I can live with that.

Thank you for your positive feedback. The homogeneity we highlighted here (Line 137-139) is for the nonlinear nature of the ENSO impact on global economy, rather than the “magnitude”. The magnitude of response to ENSO varies across countries (**Fig. 1**).

My remaining comments are as follows:

- are the results sensitive to omitting some years from the data? For instance, the authors could do a variant of rolling- or expanding-window regressions. Say, run the regression using 1960-2000 data; then 1961-2001 data, and so on. More broadly, the authors may consider some variant of cross-validation -- drop a bunch of random years, re-estimate the coefficients, repeat the process many times, and present the distribution of the coefficients.

Great point. We have tested the sensitivity of our regression model by re-estimating in running-window periods of 1960-2009, 1961-2010, ..., and 1970-2019. A similar nonlinear relationship holds across windows (**Fig. R1A-a**). We also have applied a 1000-time bootstrap test by randomly dropping 3 individual years from the 1960-2019 period, and then re-estimated our model; uncertainty exists as expected but the nature of the nonlinearity stands (**Fig. R1A-b**), and the uncertainty range is comparable to previous Bootstrap tests for re-sampling by countries, years, or 5-year blocks (**Supplementary Fig. S3**).

We have now added these sensitivity test results to **Supplementary Fig. S4**.

Fig. R1A. Sensitivity test of nonlinear effect of ENSO on global economy for (a) re-estimating by the running-window periods of 1960-2009, 1961-2010, ..., and 1970-2019 (colored curves); (b) re-estimating by randomly dropping 3 individual years from 1960-2019 based on the

Bootstrap method. Black curve indicates the original relationship. Shading indicates the 95% confidence level from the Bootstrap test.

- are the results robust to falsification checks? For instance, use the future ENSO data. Say, regress growth in period t on ENSO in periods ($t+5, t+4$, and $t+3$). There should be no statistically significant coefficients associated with ENSO variables.

Good point. We have applied a falsification check by regressing growth at year t onto future Niño3.4 index at year $t+3$, $t+4$, and $t+5$. As can be seen from **Table R1A**, there's no statistically significant α_1 and α_2 .

Table R1A. Regression coefficients for α_1 and α_2 when regressing growth onto future Niño3.4 index ($t+3$, $t+4$, and $t+5$). All coefficients have no statistically significant on 90% confidence level.

	t+3	t+4	t+5
α_1	-6.96×10^{-4}	-0.0012	-2.50×10^{-4}
α_2	6.30×10^{-4}	6.71×10^{-4}	-2.59×10^{-4}

Response to Reviewer #2

I think the paper has improved and that some of my comments have been addressed rather convincingly by the authors. However, in my reading, a few outstanding issues remain, perhaps regarding my most serious concerns.

We thank the reviewer for the positive feedback and further comments.

First. Despite some efforts at amending a few lines of the text, I feel the interpretation of results is still poor. Specifically, it is unclear what mechanism may drive the ENSO-related losses.

We thank the reviewer for delving into a detailed mechanism. There are many previous studies which provided interpretation of how ENSO drives direct losses, whereby triggering cross-sector and cross-border impact on macroeconomy.

We have provided more information in the revised manuscript (see Line 144-157), which reads as:

“... Initially, El Niño drives direct losses in the severely-affected regions through extreme weathers as reflected in temperature and precipitation anomalies. Subsequently, cross-sector and cross-border spillovers occur, affecting global macroeconomy. Previous studies suggested the cascading effects may commence via several transmission channels. For example, extreme weathers lower crop yields and agricultural productivity, leading to food shortages, trade contractions and commodity price increases (Schlenker & Roberts, 2009); abnormal sea surface temperature and ocean current cause decreased fishery stocks and other marine resources (Lehodey et al. 2006); damages on infrastructure increase reconstruction and maintenance costs and disrupt transportation networks (Kousky, 2014); fluctuation in rainfall and surface run-off cause hydroelectric power shortages and reduce water-dependent industrial outputs (Hamlet & Lettenmaier, 2007); changes in disease vector dynamics caused by extreme weathers increase healthcare cost and reduce manual productivity (Hales et al. 1999); and poor weather conditions reduce tourist arrivals and consumptions for tourism-dependent regions(Scott et al. 2012).”

- Schlenker, W., & Roberts, M. J. (2009). *Nonlinear temperature effects indicate severe damages to US crop yields under climate change. Proceedings of the National Academy of Sciences, 106(37), 15594-15598.*
- Lehodey, P., Alheit, J., Barange, M., Baumgartner, T., Beaugrand, G., Drinkwater, K., ... & Werner, F. (2006). *Climate variability, fish, and fisheries. Journal of Climate, 19(20), 5009-5030.*
- Kousky, C. (2014). *Informing climate adaptation: A review of the economic costs of natural disasters. Energy Economics, 46, 576-592.*
- Hamlet, A. F., & Lettenmaier, D. P. (2007). *Effects of 20th century warming and climate variability on flood risk in the western U.S. Water Resources Research, 43(6).*
- Hales, S., Weinstein, P., Soares, Y. & Woodward, A. *El Niño and the dynamics of vectorborne disease transmission. Environmental Health Perspectives 107, 99-102 (1999).*
- Scott, D., Gössling, S., & Hall, C. M. (2012). *Tourism and climate change: Impacts, adaptation and mitigation. Routledge.*

My reading of the main econometric model of this study (Equation 1) is that it assumes and tests how GDP growth is affected by (i) an ENSO index and (iii) local climate/weather, beyond a series of controlling factors. Results show that ENSO exerts a non-linear effect on growth conditional on local weather conditions. I wonder how. The effect cannot go through changes in yearly temperature and precipitations – as sometimes seems to emerge from the text (e.g. “As the economic impact of ENSO is underpinned by the direct climate response to ENSO for individual countries”) since they are accounted for in the econometric model. This means that El Nino and La Nina induce impacts on the economy through other channels that are relevant to growth. For example, it might well be that one of these channels is an increase in wildfires (as in the example the authors use in their response to my comment) yet only conditional on the fact that yearly precipitation and temperatures do not already reflect such an increase. However, fires are just one among many possible candidates. Interpreting the effect is even more relevant as the authors find a little role for teleconnections.

We should have made it clearer that by construction of our model, ENSO exerts impacts through anomalous temperatures/precipitations (T/P).

In our model, the annual T/P values DO NOT represent ENSO impact, but reflect slower evolution and are independent of the ENSO index.

ENSO effect starts through *anomalous* T/P (bushfire case is a good example), and via the transmission channels discussed above, triggers cross-sector spillover in affected regions and cross-border trades outside the affected regions.

The ENSO impact, as represented by the ENSO index in our model, includes the direct losses and the cascading effects.

For example, the 2020 Australia fires burned at least 19 million hectares, destroyed 2,400 buildings and killed 34 human lives and more than one billion animals. The reduced output of Australia farming sector alone amounted to about AU\$5.0B. Spill-overs to broader economies occur as decreased employment in the farming sector reduced demand, and the reduced production put upward pressure on consumer prices affecting consumers' confidence. The fires also damaged the broader economy through air pollution and harmed tourism, as the smoke haze-induced air pollution reduced worker productivity, and increased road closures and uncertainty about safety. Investment decisions are delayed. A series of cascading effects ensue. The impacts propagate outside the country through global trades and commodity prices.

Further, in response to the authors' reply to my comment, I feel the present analysis cannot say whether the macroeconomic effects are due to sectoral spillovers/cascading effects or because the direct effect of ENSO is sufficiently large in certain regions to reflect on the macroeconomic statistics. However, I appreciate such further reasoning, which I think should be further expanded to convince the reader about the large magnitude of the effects.

It is not a trivial task to quantify the direct effect and cascading effect, but we can gauge by applying the same econometric approach (Eq. 1) to different sectors of the global economy, allowing various lagged effects, By doing so, allowing contemporaneous effect only, we find

that impact on agricultural production (principally reflect the direct growth reduction) is clear, but there is a large uncertainty for non-agricultural production such as industry and service (**Fig. R2A-a**); however, in a model version allowing lagged effect (**Fig. R2A-b**), reduction of non-agricultural sectors increases in the subsequent years, with an overall nonlinear effect somewhat greater than agricultural sector, indicating a lagged but cross-sector spillover effect that is greater than the initial shock. Given that agriculture production is a small part of the global economy, it follows that the cascading effect is substantial.

Fig. R2A. Global nonlinear relationship between $D(0)JF(1)$ Niño3.4 index (normalized) and change in log GDP per capita in (a) ENSO occurrence year (year 0) and (b) 3-year cumulative (from year 0 to year 3) for different sectors of economy including agriculture (red), non-agriculture (black), industry (blue), and service (green). Shadings indicate the 95% confidence level based on a Bootstrap method.

Finally, on this point, I feel the authors should justify why an inverted U shape for the ENSO effect is reasonable. Specifically, why should an increase in the DJF Niño3.4 index be beneficial to growth up to a certain level? What is such a level? Is it reasonable? Is it robust across specifications? The figures provide little information as the ENSO index is normalized.

A positive increase in the Niño3.4 index is detrimental to growth (there seems to be a misunderstanding).

However, A moderate La Niña could benefit economic growth, as can be seen from **Fig. 1a**. La Niña events are typically associated with increased precipitation in regions including Western Pacific, part of South America and Southern Africa. Higher rainfall increases soil moisture and improve growing conditions, which boost agricultural yields and benefit the overall economy. It also leads to a higher water level in surface runoffs which improves water availability for hydroelectric power generation and human consumption (e.g., Ropelewski & Halpert, 1987; Cane et al. 1994). But La Niña also causes droughts in many other regions including America, Chile, and Peru, which would offset the benefit.

But when a La Niña is extreme, intense precipitation causes huge floods that destroy crops reducing agricultural output, destroy infrastructures, and cause life and property losses (e.g., the record-breaking floods in China derived in the 1998-99 extreme La Niña). In addition,

associated droughts in other regions added to the losses. So, large La Niña slows economic growth, contributing to the U shape effect.

We have added another x-axis by converting to an unnormalized Niño3.4 index (in degree C, **Fig. R2B**).

- Ropelewski, C. F., & Halpert, M. S. (1987). *Global and regional scale precipitation patterns associated with the El Niño/Southern Oscillation*. *Monthly Weather Review*, 115(8), 1606-1626.

- Cane, M. A., Eshel, G., & Buckland, R. W. (1994). *Forecasting Zimbabwean maize yield using eastern equatorial Pacific sea surface temperature*. *Nature*, 370(6486), 204-205.

Fig. R2B. Same as Fig. 1a, but with additional x-axis of unnormalized $D(0)JF(1)$ Niño3.4 index ($^{\circ}C$) in the top.

Second, I still have issues with the persistence of the effect the authors find. I fully agree that the impact they find in the analysis is persistent, adverse to growth in most cases, and going beyond a temporary shift in the level of output. This was motivating me to ask why they report losses in levels, which I think is misleading. Indeed, the results seem to indicate that, on average, a change in the ENSO index will lower the average growth rate of GDP, with the economy taking a few years to adjust to the new long-term growth pattern. I fear the authors tend to interpret their econometric model as if it would estimate an impulse-response effect, but this is not the case if my reading is correct. In short, I am sympathetic to the growth-effect view expressed by the authors, while I think there is an excessive and incoherent effort at expressing “damages” in lost trillions in a certain number of years (which is reminiscent of a level effect).

The reviewer is right. An estimated growth effect (reduction of GDP growth rate) was missing in terms of historical extreme El Niño events. In the revised version, we show that 1982-83, 1997-98 and 2015-16 El Niño events decreased growth by 1.0% in the occurrent year, but more than 5.0% cumulated over subsequent three years (**Fig. R2C**; Line 159-160). We presented

cumulative in-dollar-value loss, which is translated from the cumulative in-percentage growth effect, as to efficiently compare with previous estimates.

Fig. R2C. Cumulative growth effect of ENSO on global economic production in terms of $D(0)JF(1)$ Niño3.4 at each year. Red and black lines indicate the growth effect only in the ENSO occurrence year and cumulated after subsequent three years, respectively. Shadings show the 95% confidence interval based on a Bootstrap method. Extreme El Niño ($Ni\tilde{no}3.4 > 1.5$ s.d.) and La Niña ($Ni\tilde{no}3.4 < -1.25$ s.d.) events are marked as red and blue bars, respectively.

Finally, let me flag three minor points.

- I feel a figure showing the geographical distribution and intensity of losses is worth space in the main text. I suggest discussing more/including Fig. R2B or similar in the main text.

We agree with the reviewer regarding the potential interest. We have now discussed more about our country-specific approach in the main text and Methods (Line 186-193); we have also included **Supplementary Fig. S8** for detailed information.

- As far as the projection exercise is concerned, what growth rate of output (g_t) is used for future time periods (eq. 6).? Further, I still feel the difference between the projected GDP in a certain SSP and the historical growth rate of a country is NOT accounted for in the projections.

g_t refers to the GDP growth rate at year t based on the original output of SSP database.

We did not compare with the historical growth rate; instead, we compared with two futures with same ENSO evolution, one with increased ENSO amplitude but one with the same ENSO amplitude as in the model historical period.

- Please clarify exactly how the ENSO signal has been removed from country-specific temperature and precipitations for the robustness exercise. Table R2B is relevant; it is not clear how this information is used in the text.

We removed ENSO signal from country-level annual temperature and precipitation by linear regression, which goes

$$T_{it}^* = T_{it} - E_t \cdot r_i(T_i, E)$$

where T_{it}^* is the annual temperature after ENSO signal removal in country i and year t . $r_i(T_i, E)$ is the linear regression coefficient of annual temperature T_i onto the DJF Niño3.4 index E in country i during the period of 1950-2021. Same process was applied for annual precipitation. Related discussions are now included in the Method (Line 520-527).

Response to Reviewer #3

I would like to extend my appreciation to the authors for the considerable enhancements made to the paper and for addressing earlier remarks. However, there remain a few aspects that necessitate further elucidation.

Thank you for your further comments and positive feedback.

1 - The initial clarification pertains to the insensitivity of the results with respect to the ψ threshold parameter. It would be valuable to understand the authors' interpretation of this seemingly paradoxical finding.

The insensitivity of the response function can largely be attributed to the fact that the samples of countries do not differ substantially under different thresholds of ψ . There are 75 countries with $\psi > 0.5$, 63 countries with $\psi > 1$, and 50 countries with $\psi > 1.5$ (Fig. R3A). The majority of teleconnected countries fall into these three thresholds.

Fig. R3A. Distribution of country-specific ENSO teleconnection strength, with the shading shows the samples under different ψ thresholds of $\psi > 0.5$, $\psi > 1$, and $\psi > 1.5$.

2 - The explanation regarding the weighting applied to meteorological variables (based on population density) could be more explicit.

We thank the reviewer for the suggestion. There is advantage of population density weighting compared to other approaches such as a simple areal weighting. One such advantage is a better representation of the reaction of human-based economic activity to climate anomaly. Economic impact transmitted from climate shock is generally driven by human interactions and transactions. However, as we have shown, a common shock across most countries dominates, rendering our results insensitive to approaches adopted.

We have now added the detailed discussion in Method (Line 478-484).

The estimated results remain stable across various classifications; however, the only notable distinction is tied to the proportion of agriculture in GDP (low/high). The authors should delve deeper into this outcome and provide a clearer interpretation.

It is the nature of “nonlinearity” that remains stable but the amplitude of the impact is rather different. For example, the impact (in terms of growth rate) is approximately twice as large for strongly teleconnected countries as for weakly-teleconnected countries, for a +2 SD ENSO, see **Fig. 1b**.

The common nonlinear nature reflects the feature that the global spillovers and cascading effects dominate the ENSO global economic impact. On the other hand, the heterogeneous magnitude of the response is due to several different factors. One is linked to geographical locations in that strongly-teleconnected countries tend to be in the tropics and poor countries; another is economic structure in that there tends to be a high proportion of agriculture in GDP in poor countries; further, in poor countries, the level of preparedness and the capacity to mitigate are low. We have added discussion in the revised manuscript. See Line 137-139 and 610-617.

Identifying an economic cost without pinpointing the transmission channel seems problematic, particularly in a projection exercise context.

We thank the reviewer for emphasizing transmission channels. Previous studies have identified these channels. There is no suggestion that these will substantially change in the future. We have provided more information in the revised manuscript (see Line 144-157), which reads as:

“... Initially, El Niño drives direct losses in the severely-affected regions through extreme weathers as reflected in temperature and precipitation anomalies. Subsequently, cross-sector and cross-border spillovers occur, affecting global macroeconomy. Previous studies suggested the cascading effects may commence via several transmission channels. For example, extreme weathers lower crop yields and agricultural productivity, leading to food shortages, trade contractions and commodity price increases (Schlenker & Roberts, 2009); abnormal sea surface temperature and ocean current cause decreased fishery stocks and other marine resources (Lehodey et al. 2006); damages on infrastructure increase reconstruction and maintenance costs and disrupt transportation networks (Kousky, 2014); fluctuation in rainfall and surface run-off cause hydroelectric power shortages and reduce water-dependent industrial outputs (Hamlet & Lettenmaier, 2007); changes in disease vector dynamics caused by extreme weathers increase healthcare cost and reduce manual productivity (Hales et al. 1999); and poor weather conditions reduce tourist arrivals and consumptions for tourism-dependent regions(Scott et al. 2012).”

- Schlenker, W., & Roberts, M. J. (2009). *Nonlinear temperature effects indicate severe damages to US crop yields under climate change. Proceedings of the National Academy of Sciences, 106(37), 15594-15598.*

- Lehodey, P., Alheit, J., Barange, M., Baumgartner, T., Beaugrand, G., Drinkwater, K., ... & Werner, F. (2006). *Climate variability, fish, and fisheries. Journal of Climate, 19(20), 5009-5030.*

- Kousky, C. (2014). *Informing climate adaptation: A review of the economic costs of natural disasters. Energy Economics, 46, 576-592.*

- Hamlet, A. F., & Lettenmaier, D. P. (2007). *Effects of 20th century warming and climate variability on flood risk in the western U.S. Water Resources Research, 43(6).*

- Hales, S., Weinstein, P., Soares, Y. & Woodward, A. *El Niño and the dynamics of vectorborne disease transmission. Environmental Health Perspectives 107, 99-102 (1999).*

- Scott, D., Gössling, S., & Hall, C. M. (2012). *Tourism and climate change: Impacts, adaptation and mitigation. Routledge.*

3 - It is crucial to confirm that the obtained results do not rely on the population-based weighting method. For instance, I would be interested to see if the estimated costs associated with the 2015/2016 El Niño event affect China (despite its weak teleconnection) and the United States similarly when using an alternative weighting approach.

Following your suggestion, we re-estimated the country-level economic loss by using alternative weighting approaches, for example, area-weighted ENSO teleconnection, under a country-specific framework. The economic cost of 2015-16 El Niño event amounts to US\$507 and US\$912 billion for China and US under the population-weighted approach (**Fig. R3B-a**), and US\$520 and US\$926 under the area-weighted approach (**Fig. R3B-b**). The slight difference means that spillovers and cascading effects dominate.

Fig. R3B. Country-level cumulative losses from 2015-16 extreme El Niño by fitting the country-specific econometric model under (a) population-weighted teleconnection and (b) area-weighted teleconnection. The maps were created in the MATLAB computing environment using the M_Map mapping package (Pawlowicz, R., 2020. "M_Map: A mapping package for MATLAB", version 1.4m, [Computer software], available online at www.coas.ubc.ca/~rich/map.html).

4 - The authors have satisfactorily addressed the matter of ENSO temporal stability. It is worth noting, however, that the literature indicates that the occurrence of ENSO CP, EP, and mixed phenomena can considerably influence global temperatures and precipitation. My primary concern was that literature projections suggest a potentially higher frequency of CP and mixed-type events, which would have varying impacts on temperatures and precipitation.

The SST anomaly center tends to locate in the equatorial eastern Pacific during strong El Niño (EP), mostly captured by Niño3, but in central Pacific during La Niña or moderate El Niño, mostly captured by Niño4. A combination index of Niño3.4 has been a common metrics to describe all ENSO events. Niño3.4 SST variability simulated by latest models (as used by our study) captures well the overall ENSO variability of all types (Cai et al. 2022), including the mixed type of events.

- Cai, W., Ng, B., Wang, G., Santoso, A., Wu, L., & Yang, K. (2022). Increased ENSO sea surface temperature variability under four IPCC emission scenarios. *Nature Climate Change*, 12(3), 228-231.

5 - The provided response to the inquiry about introducing control variables appears inadequate. Indeed, individual fixed effects only account for the impact of time-invariant factors, and incorporating variables such as trade, the proportion of agriculture, development, and financial

depth is essential, as it could modify the estimated coefficients associated with ENSO effects. Failing to do so poses a significant risk of omitted variable bias.

We thank the reviewer for emphasizing this point. There are three kinds of control variables: country-fixed factor, year-fixed factor, and country-specific time-varying factor. Consistent with previous literatures with respect to ENSO's socio-economic impact (e.g., Hsiang et al. 2011; Smith & Ubilava, 2017), we included the country-fixed effect (μ_i) to represent time-invariant factors (e.g., geographic location, agricultural country or not), as well as the country-specific linear and quadratic time trend ($\theta_{1i}t + \theta_{2i}t^2$) for factors that change over time within a country (e.g., development, trade liberalization, technological progress).

We excluded year-fixed effect for the two reasons:

- Year-fixed effect introduces risk of collinearity as ENSO time-fixed effects could be correlated with time-specific factors, making it harder to disentangle the separate impacts from ENSO and time-fixed effects on economic growth.
- ENSO is a time-specific global phenomenon, like other major events (e.g., financial crisis) that commonly affect countries. Including year-fixed effects weakens the statistical influence of ENSO, leading to an underestimation of the real impact of ENSO on economic growth (Dell et al. 2014).

We have added detailed interpretation in Methods (Line 499-510).

- Hsiang, S. M., Meng, K. C., & Cane, M. A. (2011). Civil conflicts are associated with the global climate. *Nature*, 476(7361), 438-441.
- Smith, S. C., & Ubilava, D. (2017). The El Niño Southern Oscillation and economic growth in the developing world. *Global Environmental Change*, 45, 151-164.
- Dell, M., Jones, B. F., & Olken, B. A. (2014). What do we learn from the weather? The new climate-economy literature. *Journal of Economic literature*, 52(3), 740-798.

REVIEWER COMMENTS

Reviewer #1 (Remarks to the Author):

No further comments

Reviewer #2 (Remarks to the Author):

I would like to thank the authors for their efforts. While some of my doubts have been clarified, especially regarding the projection exercise, the two major concerns I have are not solved and, honestly, I doubt I can be fully convinced at this stage.

The authors have replied to my previous comment that (i) "ENSO exerts impacts through anomalous temperatures/precipitations (T/P)" and (ii) "annual temperature and precipitations do not represent ENSO impact, but reflect slower evolution and are independent of ENSO". If this is the authors' interpretation, it means that ENSO captures some effects on growth that local annual temperature and precipitations do not capture. First, it is not clear what such effects are; the example of fires is a good one; but beyond that, the authors are not very specific nor precise; this could be OK, but I expect some words of caution; Second, in the panel model estimated by the authors, the coefficients of temperature and precipitations can be interpreted as marginal impacts of deviations of T and P from their long-run average. Hence, they actually capture such kinds of anomalies. Third, and relatedly, the assumption that ENSO affects anomalous T/P but is independent of their first moments (or sum; which is exactly what the authors use in the estimation) is extremely strong. I would suggest the authors show, at the very least, that they are poorly correlated in key countries where the effect of El Niño is known to be sizable.

Second, I was mentioning that the estimates obtained in the paper suggest that, at least for some countries, an increase in ENSO leads to an increase in growth; and I was asking for an interpretation. The authors replied by stating that "A positive increase in the Niño3.4 index is detrimental to growth (there seems to be a misunderstanding)." and by adding Figure R2B. I think that such a Figure completely points to the fact that my interpretation is correct. First, if the estimated model is a reverse-U quadratic relationship, there must be some marginal impact that is positive and some which is negative. This is by construction. The point is whether positive marginal impacts are estimated for a relevant support of the distribution. Fig. R2B shows that, for ENSO index going from about -2.5 °C to about -1 °C, the marginal effects are positive. They may be not significant, but this is difficult to see from that figure. Relatedly, I suggest to double check that the top x-axis is correct.

Finally, I think it is important for the authors to discuss their results in light of the most recent development. I signal this relevant paper appeared in Science a few weeks ago and seems extremely relevant: Callahan, C. W., & Mankin, J. S. (2023). Persistent effect of El Niño on global economic growth. *Science*, 380(6649), 1064-1069.

Reviewer #3 (Remarks to the Author):

I would like to thank the authors for the clarifications provided. However, I believe one of my comments was not understood, and I apologize if it lacked clarity.

The authors are absolutely right to test their specifications with and without time and individual fixed effects. The inclusion of individual fixed effects, in particular, poses a risk of multicollinearity affecting the estimated coefficient attached to the main effect of ENSO. My comment was related to the inclusion of control variables explicitly in the model. Indeed, the authors include a specific linear and quadratic time trend ($\theta_1 t + \theta_2 t^2$) for factors that change over time within a country (e.g.,

development, trade liberalization, technological progress). To test the robustness of the results, would it be possible to include variables such as trade openness, the share of agricultural GDP in the overall GDP, or the depth of the financial sector? These variables vary from year to year, and their inclusion would eliminate any risk of omitted variable bias if the estimated coefficients on the variables of interest remain stable.

I have no further comments apart from this, and I would like to thank the authors for all the clarifications provided and the work done during this review process.

Response to Reviewer #1

Thank you for accepting the paper.

Response to Reviewer #2

I would like to thank the authors for their efforts. While some of my doubts have been clarified, especially regarding the projection exercise, the two major concerns I have are not solved and, honestly, I doubt I can be fully convinced at this stage.

The authors have replied to my previous comment that (i) “ENSO exerts impacts through anomalous temperatures/precipitations (T/P)” and (ii) “annual temperature and precipitations do not represent ENSO impact, but reflect slower evolution and are independent of ENSO”. If this is the authors’ interpretation, it means that ENSO captures some effects on growth that local annual temperature and precipitations do not capture.

-First, it is not clear what such effects are; the example of fires is a good one; but beyond that, the authors are not very specific nor precise; this could be OK, but I expect some words of caution;

You have a valid point. After carefully considering your point, we have now removed ENSO-induced T/P anomaly from annual T/P before incorporating into the model. On this basis, we regenerate all the results in our study.

We find that there is little difference from the result of the original approach. For example, the estimated loss from the observed extreme El Niño remains virtually the same (Fig. 2a), at US\$1.3T, 2.1T, 3.9T for the 1982/83, 1997/98, and the 2015/16 events, respectively, compared to the previous values of US\$1.3T, 2.1T, 4.0T (Line 166-167).

The small difference means that in the original approach, ENSO’s impact is, by and large, not captured by annual mean T/P but exerted through contemporaneous T/P anomaly and then via spill-over and cascading effect into broader economy in various transmission pathways (discussed in Line 145-158).

In the latest version of the revised manuscript, results are presented based on this new approach.

-Second, in the panel model estimated by the authors, the coefficients of temperature and precipitations can be interpreted as marginal impacts of deviations of T and P from their long-run average. Hence, they actually capture such kinds of anomalies.

In the new approach, ENSO is removed from annual T/P, therefore the interpretation no longer applies.

-Third, and relatedly, the assumption that ENSO affects anomalous T/P put is independent of their first moments (or sum; which is exactly what the authors use in the estimation) is extremely strong. I would suggest the authors show, at the very least, that they are poorly correlated in key countries where the effect of El Niño is known to be sizable.

Correlation of ENSO index with annual T/P in the top 10 teleconnected countries is less than 0.4, and with precipitation less than 0.2.

The low correlation is associated with a transition from El Niño to La Niña, in which anomalies during a mature El Niño phase (January, February, March) are offset by anomalies of opposite-sign during the close-to-mature La Niña phase (October to December).

The relatively small correlation underpins the small difference of the new approach from the original approach.

Second, I was mentioning that the estimates obtained in the paper suggest that, at least for some countries, an increase in ENSO leads to an increase in growth; and I was asking for an interpretation. The authors replied by stating that “A positive increase in the Niño3.4 index is detrimental to growth (there seems to be a misunderstanding).” and by adding Figure R2B. I think that such a Figure completely points to the fact that my interpretation is correct. First, if the estimated model is a reverse-U quadratic relationship, there must be some marginal impact that is positive and some which is negative. This is by construction. The point is whether positive marginal impacts are estimated for a relevant support of the distribution. Fig. R2B shows that, for ENSO index going from about -2.5 °C to about -1 °C, the marginal effects are positive. They may be not significant, but this is difficult to see from that figure. Relatedly, I suggest to double check that the top x-axis is correct.

We apologize for lack of clarity in our previous statement “A positive increase in the Niño3.4 index is detrimental to growth”. The statement only applies to the El Niño regime.

For La Niña, your expectation is true that there can be a positive marginal effect from a greater negative Niño3.4 value to a smaller negative Niño3.4 value. As we highlighted in the previous response, extreme La Niña damages but moderate La Niña benefits global economy.

The top and bottom x-axis of previous Figure R2B look very similar because the standard deviation of unnormalized Niño3.4 is close to 1 (1.0045) in the observation.

Finally, I think it is important for the authors to discuss their results in light of the most recent development. I signal this relevant paper appeared in Science a few weeks ago and seems extremely relevant: Callahan, C. W., & Mankin, J. S. (2023). Persistent effect of El Niño on global economic growth. *Science*, 380(6649), 1064-1069.

Callahan et al. use a linear but country-specific model to estimate ENSO’s economic impact. We take comfort in seeing that the findings from the two independent studies are, by and large, consistent. For example, they also found an increased loss after the initial shock, leading to an economic loss from the observed extreme El Niños that amounts to trillions of dollars globally for each event.

However, one of the differences is that their linear model assumes that La Niña/El Niño have a perfectly symmetric impact, which causes large uncertainty arising from sequences of El

Niño and La Niña events. In our nonlinear model, a strongly asymmetric impact between El Niño and La Niña leads to a persistent loss over ENSO cycles, with little dependence on ENSO sequences.

We have added discussion in Line 283-286.

Response to Reviewer #3

I would like to thank the authors for the clarifications provided. However, I believe one of my comments was not understood, and I apologize if it lacked clarity.

The authors are absolutely right to test their specifications with and without time and individual fixed effects. The inclusion of individual fixed effects, in particular, poses a risk of multicollinearity affecting the estimated coefficient attached to the main effect of ENSO. My comment was related to the inclusion of control variables explicitly in the model. Indeed, the authors include a specific linear and quadratic time trend ($\theta_{1i}t + \theta_{2i}t^2$) for factors that change over time within a country (e.g., development, trade liberalization, technological progress). To test the robustness of the results, would it be possible to include variables such as trade openness, the share of agricultural GDP in the overall GDP, or the depth of the financial sector? These variables vary from year to year, and their inclusion would eliminate any risk of omitted variable bias if the estimated coefficients on the variables of interest remain stable.

We thank the reviewer for this suggestion. It is possible to incorporate the trade openness (TO_{it}), share of agricultural GDP (SA_{it}), and financial depth (FD_{it}) as three specific control variables. For exchange, the time trend term $\theta_{1i}t + \theta_{2i}t^2$ has to be removed to avoid collinearity. The new model will be:

$$\Delta \log(y_{it}) = \sum_{l=0}^n \left\{ \alpha_{1,l} E_{t-l} + \alpha_{2,l} E_{t-l}^2 + \beta_{1,l} T_{t-l} + \beta_{2,l} T_{t-l}^2 + \lambda_{1,l} P_{t-l} + \lambda_{2,l} P_{t-l}^2 \right\} + TO_{it} + SA_{it} + FD_{it} + \mu_i + \varepsilon_{it}$$

The three variables do have some impact on economic growth (**Table R3A**). However, their inclusion appears to have little consequential impact on our result; for example, upon inclusion of them, the nonlinear function changes little (**Fig. R3A**), highlighting the robustness of our results. We have added a brief discussion in Line 510-512.

Fig. R3A. Comparison of nonlinear relationship between original model (black) and model with trade openness (TO_{it}), share of agricultural GDP (SA_{it}), and financial depth (FD_{it}).

Table R3A. Comparison of regression coefficients in the original econometric model (left) and the econometric model incorporating trade openness (TO_{it}), share of agricultural GDP (SA_{it}), and financial depth (FD_{it}) (right).

	Original, including linear and quadratic time trend	Only including trade openness, share of agricultural GDP, and financial depth
TO	-	$3.28 \times 10^{-4}***$
SA	-	$-1.88 \times 10^{-4}*$
FD	-	$-3.68 \times 10^{-4}***$
$\alpha_{1,0}$	-0.0051***	-0.0043***
$\alpha_{1,1}$	-0.0029***	-0.0026***
$\alpha_{1,2}$	-3.41×10^{-4}	1.98×10^{-4}
$\alpha_{1,3}$	-8.25×10^{-4}	-0.0014**
$\alpha_{2,0}$	3.08×10^{-4}	4.18×10^{-5}
$\alpha_{2,1}$	-0.0021***	-0.0020***
$\alpha_{2,2}$	-0.0026***	-0.0034***
$\alpha_{2,3}$	-0.0012**	-0.0011**
R^2	0.2040	0.1418

I have no further comments apart from this, and I would like to thank the authors for all the clarifications provided and the work done during this review process.

Thank you for your helpful comments on our work.

REVIEWERS' COMMENTS

Reviewer #2 (Remarks to the Author):

I would like to thank the authors for having considered and addressed all my doubts.

Reviewer #3 (Remarks to the Author):

I have no further remarks regarding this paper. I appreciate the revisions made by the authors, which I believe have improved an already high-quality initial submission. I would like to thank them for considering and responding to the various comments.

Response to Reviewer #2

Reviewer #2 (Remarks to the Author):

I would like to thank the authors for having considered and addressed all my doubts.

Thank you for accepting the paper!

Response to Reviewer #3

Reviewer #3 (Remarks to the Author):

I have no further remarks regarding this paper. I appreciate the revisions made by the authors, which I believe have improved an already high-quality initial submission. I would like to thank them for considering and responding to the various comments.

Thank you for accepting the paper!